# A Critique of Self-Expressive Deep Subspace Clustering

**Benjamin D. Haeffele**
Mathematical Institute for Data Science
Johns Hopkins University
Baltimore, MD, USA
bhaeffele@jhu.edu

**Chong You**
Department of Electrical Engineering and Computer Sciences
University of California, Berkeley
Berkeley, CA, USA
cyou@berkeley.edu

**René Vidal**
Department of Biomedical Engineering
Johns Hopkins University
Baltimore, MD, USA

## Abstract

Subspace clustering is an unsupervised clustering technique designed to cluster data that is supported on a union of linear subspaces, with each subspace defining a cluster with dimension lower than the ambient space. Many existing formulations for this problem are based on exploiting the self-expressive property of linear subspaces, where any point within a subspace can be represented as linear combination of other points within the subspace. To extend this approach to data supported on a union of non-linear manifolds, numerous studies have proposed learning an embedding of the original data using a neural network which is regularized by a self-expressive loss function on the data in the embedded space to encourage a union of linear subspaces prior on the data in the embedded space. Here we show that there are a number of potential flaws with this approach which have not been adequately addressed in prior work. In particular, we show the model formulation is often ill-posed in that it can lead to a degenerate embedding of the data, which need not correspond to a union of subspaces at all and is poorly suited for clustering. We validate our theoretical results experimentally and also repeat prior experiments reported in the literature, where we conclude that a significant portion of the previously claimed performance benefits can be attributed to an ad-hoc post processing step rather than the deep subspace clustering model.

## 1 Introduction and Background

Subspace clustering is a classical unsupervised learning problem, where one wishes to segment a given dataset into a prescribed number of clusters, and each cluster is defined as a linear (or affine) subspace with dimension lower than the ambient space. There have been a wide variety of approaches proposed in the literature to solve this problem (Vidal et al., 2016), but a large family of state-of-the-art approaches are based on exploiting the self-expressive property of linear subspaces. That is, if a point lies in a linear subspace, then it can be represented as a linear combination of other points within the subspace. Based on this fact, a wide variety of methods have been proposed which, given a dataset $\mathbf{Z} \in \mathbb{R}^{d \times N}$ of $N$ $d$-dimensional points, find a matrix of coefficients $\mathbf{C} \in \mathbb{R}^{N \times N}$ by solving the problem:

$$\min_{\mathbf{C} \in \mathbb{R}^{N \times N}} \left\{ F(\mathbf{Z}, \mathbf{C}) \equiv \frac{1}{2} \|\mathbf{Z}\mathbf{C} - \mathbf{Z}\|_F^2 + \lambda \theta(\mathbf{C}) = \frac{1}{2} \langle \mathbf{Z}^\top \mathbf{Z}, (\mathbf{C} - \mathbf{I})(\mathbf{C} - \mathbf{I})^\top \rangle + \lambda \theta(\mathbf{C}) \right\}. \quad (1)$$

Here, the first term $\|\mathbf{Z}\mathbf{C} - \mathbf{C}\|_F^2$ captures the self-expressive property by requiring every data-point to represent itself as an approximate linear combination of other points, i.e., $\mathbf{Z}_i \approx \mathbf{Z}\mathbf{C}_i$, where $\mathbf{Z}_i$ and $\mathbf{C}_i$ are the $i^{\text{th}}$ columns of $\mathbf{Z}$ and $\mathbf{C}$, respectively. The second term, $\theta(\mathbf{C})$, is some regularization function designed to encourage each data point to only select other points within the

correct subspace in its representation and to avoid trivial solutions (such as $\mathbf{C} = \mathbf{I}$). Once the $\mathbf{C}$ matrix has been solved for, one can then define a graph affinity between data points, typically based on the magnitudes of the entries of $\mathbf{C}$, and use an appropriate graph-based clustering method (e.g., spectral clustering (von Luxburg, 2007)) to produce the final clustering of the data points.

One of the first methods to utilize this approach was Sparse Subspace Clustering (SSC) (Elhamifar & Vidal, 2009; 2013), where $\theta$ takes the form $\theta_{SSC}(\mathbf{C}) = \|\mathbf{C}\|_1 + \delta(\mathrm{diag}(\mathbf{C}) = \mathbf{0})$, with $\| \cdot \|_1$ denoting the $\ell_1$ norm and $\delta$ an indicator function which takes value $\infty$ if an element of the diagonal of $\mathbf{C}$ is non-zero and $0$ otherwise. By regularizing $\mathbf{C}$ to be sparse, a point must represent itself using the smallest number of other points within the dataset, which in turn ideally requires a point to only select other points within its own subspace in the representation. Likewise other variants, with Low-Rank Representation (LRR) (Liu et al., 2013), Low-Rank Subspace Clustering (LRSC) (Vidal & Favaro, 2014) and Elastic-net Subspace Clustering (EnSC) (You et al., 2016) being well-known examples, take the same form as (1) with different choices of regularization. For example, $\theta_{LRR}(\mathbf{C}) = \|\mathbf{C}\|_*$ and $\theta_{EnSC}(\mathbf{C}) = \|\mathbf{C}\|_1 + \tau\|\mathbf{C}\|_F^2 + \delta(\mathrm{diag}(\mathbf{C}) = \mathbf{0})$, where $\| \cdot \|_*$ denotes the nuclear norm (sum of the singular values). A significant advantage of the majority of these methods is that it can be proven (typically subject to some technical assumptions regarding the angles between the underlying subspaces and the distribution of the sampled data points within the subspaces) that the optimal $\mathbf{C}$ matrix in (1) will be "correct" in the sense that if $\mathbf{C}_{i,j}$ is non-zero then the $i$th and $j$th columns of $\mathbf{Z}$ lie in the same linear subspace (Soltanolkotabi & Candès, 2012; Lu et al., 2012; Elhamifar & Vidal, 2013; Soltanolkotabi et al., 2014; Wang et al., 2015; Wang & Xu, 2016; You & Vidal, 2015a;b; Yang et al., 2016; Tsakiris & Vidal, 2018; Li et al., 2018; You et al., 2019; Robinson et al., 2019), which has led to these methods achieving state-of-the-art performance in many applications.

## 1.1 Self-Expressive Deep Subspace Clustering

Although subspace clustering techniques based on self-expression display strong empirical performance and provide theoretical guarantees, a significant limitation of these techniques is the requirement that the underlying dataset needs to be approximately supported on a union of linear subspaces. This has led to a strong motivation to extend these techniques to more general datasets, such as data supported on a union of non-linear low-dimensional manifolds. From inspection of the right side of (1), one can observe that the only dependence on the data $\mathbf{Z}$ comes in the form of the Gram matrix $\mathbf{Z}^\top\mathbf{Z}$. As a result, self-expressive subspace clustering techniques are amendable to the "kernel-trick", where instead of taking an inner product kernel between data points, one can instead use a general kernel $\kappa(\cdot, \cdot)$ (Patel & Vidal, 2014). Of course, such an approach comes with the traditional challenge of how to select an appropriate kernel so that the embedding of the data in the Hilbert space associated with the choice of kernel results in a union of linear subspaces.

The first approach to propose learning an appropriate embedding of an initial dataset $\mathbf{X} \in \mathbb{R}^{d_x \times N}$ (which does not necessarily have a union of subspaces structure) was given by Patel et al. (2013; 2015) who proposed first projecting the data into a lower dimensional space via a learned linear projector, $\mathbf{Z} = \mathbf{P}_l\mathbf{X}$, where $\mathbf{P}_l \in \mathbb{R}^{d \times d_x}$ $(d < d_x)$ is also optimized over in addition to $\mathbf{C}$ in (1). To ensure that sufficient information about the original data $\mathbf{X}$ is preserved in the low-dimensional embedding $\mathbf{Z}$, the authors further required that the linear projector satisfy the constraint that $\mathbf{P}_l\mathbf{P}_l^\top = \mathbf{I}$ and added an additional term to the objective with form $\|\mathbf{X} - \mathbf{P}_l^\top\mathbf{P}_l\mathbf{X}\|_F^2$. However, since the projector is linear, the approach is not well suited for nonlinear manifolds, unless it is augmented with a kernel embedding, which again requires choosing a suitable kernel.

More recently, given the success of deep neural networks, a large number of studies Peng et al. (2017); Ji et al. (2017); Zeng et al. (2019b;a); Xie et al. (2020); Sun et al. (2019); Li et al. (2019); Yang et al. (2019); Jiang et al. (2019); Tang et al. (2018); Kheirandishfard et al. (2020b); Zhou et al. (2019); Jiang et al. (2018); Abavisani & Patel (2018); Zhou et al. (2018); Zhang et al. (2018; 2019b;a); Kheirandishfard et al. (2020a) have attempted to learn an appropriate embedding of the data (which ideally would have a union of linear subspaces structure) via a neural network, $\Phi_E(\mathbf{X}, \mathcal{W}_e)$, where $\mathcal{W}_e$ denotes the parameters of a network mapping defined by $\Phi_E$, which takes a dataset $\mathbf{X} \in \mathbb{R}^{d_x \times N}$ as input. In an attempt to encourage the embedding of the data, $\Phi_E(\mathbf{X}, \mathcal{W}_e)$, to have this union of subspaces structure, these approaches minimize a self-expressive loss term, with form given in (1), on the embedded data, and a large majority of these proposed techniques can be

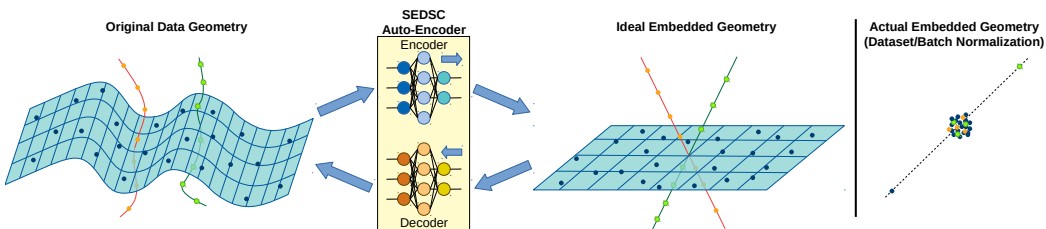

Figure 1: Illustration of our theoretical results. The goal of the SEDSC model is to train a network (typically an auto-encoder) to map data from a union of non-linear manifolds (**Left**) to a union of linear subspaces (**Center**). However, we show that for many of the formulations that have been proposed in the literature the global optimum of the model will have a degenerate geometry in the embedded space. For example, in the Dataset and Channel/Batch normalization schemes Theorem 2 shows that the globally optimal geometry will have all points clustered near the origin with the exception of two points, which will be copies of each other (to within a sign-flip) (**Right**).

described by the general form:

$$\min_{\mathcal{W}_e, \mathbf{C}} \gamma F(\mathbf{Z}, \mathbf{C}) + g(\mathbf{Z}, \mathbf{X}, \mathbf{C}) \text{ subject to } \mathbf{Z} = \Phi_E(\mathbf{X}, \mathcal{W}_e) \tag{2}$$

where $g$ is some function designed to discourage trivial solutions (for example $\Phi_E(\mathbf{X}, \mathcal{W}_e) = \mathbf{0}$) and $\gamma > 0$ is some hyper-parameter to balance the terms.

Several different choices of $g$ have been proposed in the literature. The first is to place some form of normalization directly on $\mathbf{Z}$. For example, Peng et al. (2017) propose an **Instance Normalization** regularization, $g(\mathbf{Z}, \mathbf{X}, \mathbf{C}) = \sum_{i=1}^{N} (\mathbf{Z}_i^\top \mathbf{Z}_i - 1)^2$, which attempts to constrain the norm of the embedded data points to be 1. Likewise, one could also consider **Dataset Normalization** schemes, which bound the norm of the entire embedded representation $\|\mathbf{Z}\|_F^2 \geq \tau$ or **Channel/Batch Normalization** schemes, which bound the norm of a channel of the embedded representation (i.e., a row of $\mathbf{Z}$), $\|\mathbf{Z}^i\|_F^2 \geq \tau, \forall i$. We note that this is quite similar to the common Batch Norm operator (Ioffe & Szegedy, 2015) used in neural network training which attempts to constrain each row of $\mathbf{Z}$ to have zero mean and constant norm.

Another popular form of $g$ is to also train a decoding network $\Phi_D(\cdot, \mathcal{W}_d)$ with parameters $\mathcal{W}_d$ to map the self-expressive representation, $\Phi_E(\mathbf{X}, \mathcal{W}_e)\mathbf{C}$, back to the original data to ensure that sufficient information is preserved in the self-expressive representation to recover the original data. We will refer to this as **Autoencoder Regularization**. This idea is essentially a generalization of the previously discussed work, which considered constrained linear encoder/decoder mappings (Patel et al., 2013; Patel & Vidal, 2014; Patel et al., 2015), to non-linear autoencoding neural networks and was first proposed by the authors of Ji et al. (2017). The problem takes the general form:

$$\min_{\mathcal{W}_e, \mathcal{W}_d, \mathbf{C}} \gamma F(\mathbf{Z}, \mathbf{C}) + \ell(\mathbf{X}, \Phi_D(\mathbf{ZC}, \mathcal{W}_d)) \text{ subject to } \mathbf{Z} = \Phi_E(\mathbf{X}, \mathcal{W}_e), \tag{3}$$

where the first term is the standard self-expressive subspace clustering loss applied to the embedded representation, and the second term is a standard auto-encoder loss, with $\ell$ typically chosen to be the squared loss. Note that here both the encoding/decoding network and the optimal self-expression encoding, $\mathbf{C}$, are trained jointly, and once problem (3) is solved one can use the recovered $\mathbf{C}$ matrix directly for clustering.

Using the general formulation in (2) and the popular specific case in (3), Self-Expressive Deep Subspace Clustering (SEDSC) has been applied to a variety of applications, but there is relatively little that it known about it from a theoretical standpoint. Initial formulations for SEDSC were guided by the intuition that if the dataset is drawn from a union of linear subspaces, then solving problem (1) is known to induce desirable properties in $\mathbf{C}$ for clustering. By extension one might assume that if one also optimizes over the geometry of the learned embedding ($\mathbf{Z}$) this objective might induce a desirable geometry in the embedded space (e.g., a union of linear subspaces). However, a vast majority of the prior theoretical analysis for problems of the form in (1) only considers the case where the data is held fixed and analyzes the properties of the optimal $\mathbf{C}$ matrix. Due to the well-known fact that neural networks are capable of producing highly-expressive mapping functions (and hence a network could produce many potential values for $\mathbf{Z}$), the use of a model such as (2)/(3) is essentially using (1) as a *regularization function* on $\mathbf{Z}$ to encourage a union of subspaces geometry. To

date, however, models such as (2)/(3) have been guided largely by intuition and significant questions remain regarding what type of data geometry is encouraged by $F(\mathbf{Z}, \mathbf{C})$ when one optimizes over both the encoding matrix, $\mathbf{C}$, and the network producing the embedded data representation, $\mathbf{Z}$.

## 1.2 PAPER CONTRIBUTIONS

Here we explore these questions via theoretical analysis where we show that the use of $F(\mathbf{Z}, \mathbf{C})$ as a regularization function when learning a kernel from the data in an attempt to promote beneficial data geometries in the embedded space, as is done in (2), is largely insufficient in the sense that the optimal data geometries are trivial and not conducive to successful clustering in most cases. Specifically, we first note a basic fact that the Autoencoder Regularization formulation in (3) is typically ill-posed for most commonly used networks if constraints are not placed on the magnitude of the embedded data, $\Phi_E(\mathbf{X}, \mathcal{W}_e)$, either through regularization/constraints on the autoencoder weights/architecture or an explicit normalization of $\Phi_E(\mathbf{X}, \mathcal{W}_e)$, such as in the Instance/Batch/Dataset Normalization schemes. Then, even assuming that the embedded representation has been suitably normalized, we show that the optimal embedded data geometry encouraged by $F(\mathbf{Z}, \mathbf{C})$ is trivial in various ways, which will depend on how the data is normalized. We illustrate our theoretical predictions with experiments on both real and synthetic data. Further, we show experimentally that much of the claimed performance benefit of the SEDSC model reported in previous work can be attributed to an ad-hoc post-processing of the $\mathbf{C}$ matrix first proposed in Ji et al. (2017).

**Notation.** We will denote matrices with capital boldfaced letters, $\mathbf{Z}$, vectors which are not rows/columns of a larger matrix with lower-case boldfaced letters, $\mathbf{z}$, and sets with calligraphic letters, $\mathcal{Z}$. The $i^{\text{th}}$ row of a matrix will be denoted with a superscript, $\mathbf{Z}^i$; the $i^{\text{th}}$ column of a matrix will be denoted with a subscript, $\mathbf{Z}_i$; the $(i, j)^{\text{th}}$ entry of a matrix will be denoted as $\mathbf{Z}_{i,j}$; and the $i^{\text{th}}$ entry of a vector will be denoted as $\mathbf{z}_i$. We will denote the minimum singular value of a matrix $\mathbf{Z}$ as $\sigma_{\min}(\mathbf{Z})$, and we will denote the nuclear, Frobenius, $\ell_p$, and Schatten-$p$ norms[1] for a matrix/vector as $\|\mathbf{Z}\|_*$, $\|\mathbf{Z}\|_F$, $\|\mathbf{Z}\|_p$, and $\|\mathbf{Z}\|_{\mathcal{S}_p}$ respectively. $\delta(cnd)$ denotes an indicator function with value 0 if condition $cnd$ is true and $\infty$ otherwise.

## 2 THEORETICAL ANALYSIS

### 2.1 BASIC SCALING ISSUES

We begin our analysis of the SEDSC model by considering the most popular formulation, which is to employ Autoencoder Regularization as in (3). Specifically, we note that without any regularization on the autoencoder network parameters $(\mathcal{W}_e, \mathcal{W}_d)$ or any normalization placed on the embedded representation, $\Phi_E(\mathbf{X}, \mathcal{W}_e)$, the formulation in (3) is often ill-posed in the sense that the value of $F(\Phi_E(\mathbf{X}, \mathcal{W}_e), \mathbf{C})$ can be made arbitrarily small without changing the value of the autoencoder loss by simply scaling-down the weights in the encoding network and scaling-up the weights in the decoding network in a way which doesn't change the output of the autoencoder but reduces the magnitude of the embedded representation. As we will see, a sufficient condition for this to be possible is when the non-linearity in the final layer of the encoder is positively-homogeneous[2]. We further note that most commonly used non-linearities in neural networks are positively homogenous, with Rectified Linear Units (ReLUs), leaky ReLUs, and max-pooling being common examples. As a result, most autoencoder architectures employed for SEDSC will require some form of regularization or normalization for the $F$ term to have any impact on the geometry of the embedded representation (other than trivially becoming smaller in magnitude), though many proposed formulations do not take this issue into account.

As a basic example, consider fully-connected single-hidden-layer networks for the encoder/decoder, $\Phi_E(\mathbf{X}, \mathcal{W}_e) = \mathbf{W}_e^2(\mathbf{W}_e^1\mathbf{X})_+$ and $\Phi_D(\mathbf{Z}, \mathcal{W}_d) = \mathbf{W}_d^2(\mathbf{W}_d^1\mathbf{Z})_+$, where $(x)_+ = \max\{x, 0\}$ denotes the Rectified Linear Unit (ReLU) non-linearity applied entry-wise. Then, note that because the ReLU function is positively homogeneous of degree 1 for any $\alpha \geq 0$ one has $\Phi_E(\mathbf{X}, \alpha\mathcal{W}_e) = (\alpha\mathbf{W}_e^2)(\alpha\mathbf{W}_e^1\mathbf{X})_+ = \alpha^2\mathbf{W}_e^2(\mathbf{W}_e^1\mathbf{X})_+ = \alpha^2\Phi_E(\mathbf{X}, \mathcal{W}_e)$ and similarly for $\Phi_D$. As a result one can scale $\mathcal{W}_e$ by any $\alpha > 0$ and $\mathcal{W}_d$ by $\alpha^{-1}$ without changing the output of

---

[1] Recall, the **Schatten-$p$ norm** of a matrix $\mathbf{C}$ is defined as the $\ell_p$ norm, $p \geq 1$, of the singular values of $\mathbf{C}$.

[2] Recall, a function $f(x)$ is **positively-homogeneous of degree** $p$ if $f(\alpha x) = \alpha^p f(x)$, $\forall \alpha \geq 0$.

the autoencoder, $\Phi_D(\Phi_E(\mathbf{X}, \alpha\mathcal{W}_e), \alpha^{-1}\mathcal{W}_d) = \Phi_D(\Phi_E(\mathbf{X}, \mathcal{W}_e), \mathcal{W}_d)$, but $\|\Phi_E(\mathbf{X}, \alpha\mathcal{W}_e)\|_F = \alpha^2\|\Phi_E(\mathbf{X}, \mathcal{W}_e)\|_F$. From this, taking $\alpha$ to be arbitrarily small the magnitude of the embedding can also be made arbitrarily small, so if $\theta(\mathbf{C})$ in (1) is something like a norm (as is typically the case) the value of $F(\mathbf{Z}, \mathbf{C})$ in (3) can also be made arbitrarily small without changing the reconstruction loss of the autoencoder $\ell$. This implies one is simply training an autoencoder without any contribution from the $F(\mathbf{Z}, \mathbf{C})$ term (other than trying to reduce the magnitude of the embedded representation). This basic idea can be easily generalized and formalized in the following statement (all proofs are provided in the Supplementary Material):

**Proposition 1.** *Consider the objective in* (3) *and suppose* $\theta(\mathbf{C})$ *in* (1) *is a function which achieves its minimum at* $\mathbf{C} = \mathbf{0}$ *and satisfies* $\theta(\mu\mathbf{C}) < \theta(\mathbf{C})$, $\forall\mu \in (0, 1)$. *If for any choice of* $(\mathcal{W}_d, \mathcal{W}_e)$ *and* $\tau_1, \tau_2 \in (0, 1)$ *there exists* $(\hat{\mathcal{W}}_d, \hat{\mathcal{W}}_e)$ *such that* $\Phi_E(\mathbf{X}, \hat{\mathcal{W}}_e) = \tau_1\Phi_E(\mathbf{X}, \mathcal{W}_e)$ *and* $\Phi_D(\tau_2\mathbf{Z}, \hat{\mathcal{W}}_d) = \Phi_D(\mathbf{Z}, \mathcal{W}_d)$ *then the* $F$ *term in* (3) *can be made arbitrarily small without changing the value of the loss function* $\ell$. *Further, if the final layer of the encoding network is positively-homogeneous with degree* $p \neq 0$, *such a* $(\hat{\mathcal{W}}_d, \hat{\mathcal{W}}_e)$ *will always exist simply by scaling the network weights of the linear (or affine) operator parameters.*

In practice, most of the previously cited studies employing the SEDSC model use networks which satisfy the conditions of Proposition 1 without regularization, and we note that such models can never be solved to global optimality (only approach it asymptotically) and are inherently ill-posed. From this, one can conclude that Autoencoder Regularization by itself is often insufficient to prevent trival solutions to (2). This specific issue can be easily fixed (although prior work often does not) if one ensures that the magnitude of the embedded representation is constrained to be larger than some minimum value - either through regularization/constraints placed on the network weights, such as using weight decay or coupling weights between the encoder and decoder, potentially a different choice of non-linearity on the last layer or the encoder, or explicit normalization of the embedded representation. However, the question remains what geometries are promoted by (1) even if the basic issue described above is corrected, which we explore next.

## 2.2 THE EFFECT OF THE AUTOENCODER LOSS

Before presenting the remainder of our formal results, we first pause to discuss the effect of the autoencoder loss ($\ell$) in (3). The use of the autoencoder loss is designed to ensure that the embedded representation $\mathbf{Z}$ retains sufficient information about the original data $\mathbf{X}$ so that the data can be reconstructed by the decoder, but we note that this *does not* necessarily impose significant constraints on the geometric arrangement of the embedded points in $\mathbf{Z}$. While there is potentially some constraint on the possible geometries of $\mathbf{Z}$ which is imposed by the choice of encoder/decoder architectures, reasonably expressive encoders and decoders can map a finite number of data points arbitrarily into the embedded space and still decode them accurately, provided the embedding of any two points is still distinct to within some $\epsilon$ perturbation (i.e., $\Phi_E$ is a one-to-one mapping on the points in $\mathbf{X}$). Further, because we only evaluate the mapping on a finite set of points $\mathbf{X}$, this is a much easier (and can be achieved with much simpler networks) than the well-known universal approximation regime of neural networks (which requires good approximation over the entire continuous data domain), and it is well documented that typical network architectures can optimally fit fairly arbitrary finite training data (Zhang et al., 2017).

Given the above discussion, in our analysis we will first focus on the setting where the autoencoder is highly expressive. In this regime, the encoder can select weight parameters to produce an essentially arbitrary choice of $\mathbf{Z}$ embedding, and as long as $\Phi_E(\mathbf{X}_i) \neq \Phi_E(\mathbf{X}_j), \forall\mathbf{X}_i \neq \mathbf{X}_j$ then the decoder can exactly recover the original data $\mathbf{X}$. As a result, for almost any choice of encoder the autoencoder loss term, $\ell$, in (3) can be exactly minimized, so the optimal network weights for the model in (3) (and likewise (2)) will be those which minimize $F(\mathbf{Z}, \mathbf{C})$ (potentially to within some small $\epsilon$ perturbation so that $\Phi_E(\mathbf{X})$ is a one-to-one mapping on the points in $\mathbf{X}$). As we already know from Prop 1, this is ill-posed without some additional form of regularization, so in the following subsections we explore optimal solutions to $F(\mathbf{Z}, \mathbf{C})$ when one optimizes over both $\mathbf{C}$ and the embedded representation $\mathbf{Z} = \Phi_E(\mathbf{X}, \mathcal{W}_e)$ subject to three different constraints on $\mathbf{Z}$: (1) Dataset Normalization $\|\mathbf{Z}\|_F^2 \geq \tau$, (2) Channel/Batch Normalization $\|\mathbf{Z}^i\|_F \geq \tau \, \forall i$, and (3) Instance Normalization $\|\mathbf{Z}_i\|_F = \tau \, \forall i$. Finally, after characterizing solutions to $F(\mathbf{Z}, \mathbf{C})$, we give a specific example of a very simple (i.e., not highly expressive) family of architectures which can achieve these solutions,

showing that the assumption of highly expressive networks is not necessary for these solutions to be globally optimal.

## 2.3 DATASET AND BATCH/CHANNEL NORMALIZATION

We will first consider the Dataset and Batch/Channel Normalization schemes, which will both result in very similar optimal solutions for the embedded data geometry. Recall, that this considers when the entire dataset is constrained to have a norm greater than some minimum value in the embedded space (Dataset Normalization) or when the norms of the rows of the dataset are constrained to have a norm greater than some minimum value (Batch/Channel Normalization). We note that the latter case is very closely related to batch normalization (Ioffe & Szegedy, 2015), which requires that each channel/feature (in this case a row of the embedded representation) to have zero mean and constant norm in expectation over the draw of a mini-batch. Additionally, while we do not explicitly enforce a zero-mean constraint, we will see that optimal solutions will exist which have zero mean. Now, if one optimizes $F(\mathbf{Z}, \mathbf{C})$ jointly over $(\mathbf{Z}, \mathbf{C})$ subject to the above constraint(s) on $\mathbf{Z}$, then the following holds:

**Theorem 1.** *Consider the following optimization problems which jointly optimize over $\mathbf{Z}$ and $\mathbf{C}$:*

$$\text{(P1)} \quad \min_{\mathbf{C},\mathbf{Z}} \left\{ F(\mathbf{Z}, \mathbf{C}) \text{ s.t. } \|\mathbf{Z}\|_F^2 \geq \tau \right\} \quad \text{(P2)} \quad \min_{\mathbf{C},\mathbf{Z}} \left\{ F(\mathbf{Z}, \mathbf{C}) \text{ s.t. } \|\mathbf{Z}^i\|_F^2 \geq \tfrac{\tau}{d} \ \forall i \right\} \quad (4)$$

*Then optimal values for $\mathbf{C}$ for both* (P1) *and* (P2) *are given by*

$$\mathbf{C}^* \in \arg\min_{\mathbf{C}} \tfrac{1}{2}\sigma_{\min}^2(\mathbf{C} - \mathbf{I})\tau + \lambda\theta(\mathbf{C}). \quad (5)$$

*Moreover, for any optimal $\mathbf{C}^*$, let $r$ be the multiplicity of the smallest singular value of $\mathbf{C}^* - \mathbf{I}$ and let $\mathbf{Q} \in \mathbb{R}^{N \times r}$ be an orthonormal basis for the subspace spanned by the left singular vectors associated with the smallest singular value of $\mathbf{C}^* - \mathbf{I}$. Then we have that optimal values for $\mathbf{Z}$ are given by:*

$$\mathbf{Z}^* \in \{\mathbf{B}\mathbf{Q}^\top \colon \mathbf{B} \in \mathbb{R}^{d \times r} \cap \mathcal{B}\}, \quad \mathcal{B} = \begin{cases} \begin{cases} \{\mathbf{B} : \|\mathbf{B}\|_F^2 = \tau\} & \sigma_{\min}(\mathbf{C}^* - \mathbf{I}) > 0 \\ \{\mathbf{B} : \|\mathbf{B}\|_F^2 \geq \tau\} & \sigma_{\min}(\mathbf{C}^* - \mathbf{I}) = 0 \end{cases} & \text{(P1)} \\ \begin{cases} \{\mathbf{B} : \|\mathbf{B}^i\|_F^2 = \tfrac{\tau}{d}, \forall i\} & \sigma_{\min}(\mathbf{C}^* - \mathbf{I}) > 0 \\ \{\mathbf{B} : \|\mathbf{B}^i\|_F^2 \geq \tfrac{\tau}{d}, \forall i\} & \sigma_{\min}(\mathbf{C}^* - \mathbf{I}) = 0 \end{cases} & \text{(P2)} \end{cases} \quad (6)$$

From the above result, one notes from (5) that optimizing $F(\mathbf{Z}, \mathbf{C})$ jointly over both $\mathbf{Z}$ and $\mathbf{C}$ is equivalent to finding a $\mathbf{C}$ which minimizes a trade-off between the minimum singular value of $\mathbf{C} - \mathbf{I}$ and the regularization $\theta(\mathbf{C})$. Further, we note that if such an optimal $\mathbf{C}$ results in the minimum singular value of $\mathbf{C} - \mathbf{I}$ having a multiplicity of 1, then this implies that every data point in $\mathbf{Z}$ will simply be a scaled version of the same point. Obviously, such an embedding is not useful for subspace clustering. Characterizing optimal solutions to (5) is somewhat complicated in the general case due to the fact that the smallest singular value is a concave function of a matrix and $\theta$ is typically chosen to be a convex regularization function, resulting in the minimization of a convex+concave function. Instead, we will focus on the most commonly used choices of regularization, starting with $\theta_{SSC}(\mathbf{C}) = \|\mathbf{C}\|_1 + \delta(\text{diag}(\mathbf{C}) = \mathbf{0})$, where we derive the optimal solution in the case where $\sigma_{\min}(\mathbf{C} - \mathbf{I}) = 0$. We note that this corresponds to the case where $\mathbf{Z}^* = \mathbf{Z}^*\mathbf{C}^*$ which one typically obtains as $\lambda$ in (1) is taken to be small.

**Theorem 2.** *Optimal solutions to the problems*

$$\text{(P1)} \quad \min_{\mathbf{Z},\mathbf{C}} \|\mathbf{C}\|_1 \text{ s.t. } \text{diag}(\mathbf{C}) = \mathbf{0}, \ \mathbf{Z} = \mathbf{Z}\mathbf{C}, \ \|\mathbf{Z}\|_F^2 \geq \tau \quad (7)$$

$$\text{(P2)} \quad \min_{\mathbf{Z},\mathbf{C}} \|\mathbf{C}\|_1 \text{ s.t. } \text{diag}(\mathbf{C}) = \mathbf{0}, \ \mathbf{Z} = \mathbf{Z}\mathbf{C}, \ \|\mathbf{Z}^i\|_F^2 \geq \tfrac{\tau}{d} \ \forall i \quad (8)$$

*are characterized by the set*

$$(\mathbf{Z}^*, \mathbf{C}^*) \in \left\{ \begin{bmatrix} \mathbf{z} & \mathbf{z} & \mathbf{0}_{d \times N-2} \end{bmatrix} \mathbf{P} \right\} \times \left\{ \mathbf{P}^\top \begin{bmatrix} 0 & 1 & 0 & \cdots & 0 \\ 1 & 0 & 0 & \cdots & 0 \\ 0 & 0 & 0 & \cdots & 0 \\ \vdots & \vdots & \vdots & \ddots & \\ 0 & 0 & 0 & & 0 \end{bmatrix} \mathbf{P} \right\}, \quad (9)$$

*where $\mathbf{P} \in \mathbb{R}^{N \times N}$ is an arbitrary signed-permutation matrix and $\mathbf{z} \in \mathbb{R}^d$ is an arbitrary vector which satisfies $\|\mathbf{z}\|_F^2 \geq \tau/2$ for* (P1) *and $\mathbf{z}_i^2 \geq \tau/(2d), \forall i \in [1, d]$ for* (P2).

From the above result, we have shown that if $\mathbf{Z}$ is normalized to have a lower-bounded norm on either the entire embedded representation or for each row, then the effect of the $F(\mathbf{Z}, \mathbf{C})$ loss will be largely similar to the situation described by Proposition 1 in the sense that the loss will still attempt to push all of the points to 0 with the exception of two points, which will be copies of each other (potentially to within a sign-flip). Again, the optimal embedded representation is clearly ill-posed for clustering since all but two of the points are trivially driven towards 0 in the embedded space.

In addition, we also present a result similar to Theorem 2 when $\mathbf{C}$ is regularized by any Schatten-$p$ norm, which includes two other popular choices of regularization that have appeared in the literature – the Frobenius norm $\theta(\mathbf{C}) = \|\mathbf{C}\|_F$ (for $p = 2$) and the nuclear norm $\theta(\mathbf{C}) = \|\mathbf{C}\|_*$ (for $p = 1$) – as special cases.

**Theorem 3.** *Optimal solutions to the problems*

$$\text{(P1)} \quad \min_{\mathbf{Z},\mathbf{C}} \|\mathbf{C}\|_{\mathcal{S}_p} \ \text{s.t.} \ \mathbf{Z} = \mathbf{Z}\mathbf{C}, \ \|\mathbf{Z}\|_F^2 \geq \tau \tag{10}$$

$$\text{(P2)} \quad \min_{\mathbf{Z},\mathbf{C}} \|\mathbf{C}\|_{\mathcal{S}_p} \ \text{s.t.} \ \mathbf{Z} = \mathbf{Z}\mathbf{C}, \ \|\mathbf{Z}^i\|_F^2 \geq \tfrac{\tau}{d} \ \forall i, \tag{11}$$

*where $\|\mathbf{C}\|_{\mathcal{S}_p}$ is any Schatten-$p$ norm on $\mathbf{C}$, are characterized by the set*

$$(\mathbf{Z}^*, \mathbf{C}^*) \in \left\{ (\mathbf{z}\mathbf{q}^\top) \times (\mathbf{q}\mathbf{q}^\top) : \mathbf{q} \in \mathbb{R}^N, \|\mathbf{q}\|_F = 1 \right\} \tag{12}$$

*where $\mathbf{z} \in \mathbb{R}^d$ is an arbitrary vector which satisfies $\|\mathbf{z}\|_F^2 \geq \tau$ for (P1) and $\mathbf{z}_i^2 \geq \tfrac{\tau}{d}, \ \forall i$ for (P2).*

Again, note that this is obviously not a good geometry for successful spectral clustering as all the points in the dataset are simply arranged on a single line and the optimal $\mathbf{C}$ is a rank-one matrix.

## 2.4 INSTANCE NORMALIZATION

To explicitly prevent the case where most of the points in the embedded space are trivially driven to 0 as in the prior two normalization schemes, another potential normalization strategy which has been proposed for (2) is to use Instance Normalization (Peng et al., 2017), where the $\ell_2$ norm of each embedded data point is constrained to be equal to some constant. Here again we will see that this results in somewhat trivial data geometries. Specifically, we will again focus on the choice of the $\theta_{SSC}(\mathbf{C})$ regularization function when we have exact equality, $\mathbf{Z} = \mathbf{Z}\mathbf{C}$, for simplicity of presentation. From this we have the following result:

**Theorem 4.** *Optimal solutions to the problem*

$$\min_{\mathbf{Z},\mathbf{C}} \|\mathbf{C}\|_1 \ \text{s.t.} \ \text{diag}(\mathbf{C}) = \mathbf{0}, \ \mathbf{Z} = \mathbf{Z}\mathbf{C}, \ \|\mathbf{Z}_i\|_F^2 = \tau \ \forall i \tag{13}$$

*must have the property that for any column in $\mathbf{Z}^*$, $\mathbf{Z}_i^*$, there exists another column, $\mathbf{Z}_j^*$ ($i \neq j$), such that $\mathbf{Z}_i^* = s_{i,j}\mathbf{Z}_j^*$ where $s_{i,j} \in \{-1, 1\}$. Further, $\|\mathbf{C}_i^*\|_1 = 1 \ \forall i$ and $\mathbf{C}_{i,j} \neq 0 \implies \mathbf{Z}_i^* = \pm\mathbf{Z}_j^*$.*

The above result is quite intuitive in the sense that because a given point cannot use itself in its representation (due to the $\text{diag}(\mathbf{C}) = \mathbf{0}$ constraint), the next best thing is to have an exact copy of itself in another column. While this result is more conducive to successful clustering in the sense that points which are close in the embedded space are encouraged to 'merge' into a single point, there are still numerous pathological geometries that can result. Specifically, there is no constraint on the number of 'distinct' points in the representation (i.e., the number of vectors which are not copies of each other to within a sign flip), other than it must be less than $N/2$. As a result, the optimal $\mathbf{C}^*$ matrix can also contain an arbitrary number (in the range $[1, N/2]$) of connected components in the affinity graph, resulting in somewhat arbitrary spectral clustering.

**Example of Degenerate Geometry with Simple Networks.** In section 2.2 we discussed how if the encoder/decoder are highly expressive then the optimal embedding will approach the solutions we give in our theoretical analysis. Here we show that trivial embeddings can also occur with relatively simple encoders/decoders. Specifically, consider basic encoder/decoder architectures which consist of two affine mappings with a ReLU non-linearity (denoted as $(\cdot)_+$) on the hidden layer:

$$\Phi_E(\mathbf{x}, \mathcal{W}_e) = \mathbf{W}_e^2(\mathbf{W}_e^1\mathbf{x} + \mathbf{b}_e^1)_+ + \mathbf{b}_e^2 \qquad \Phi_D(\mathbf{z}, \mathcal{W}_d) = \mathbf{W}_d^2(\mathbf{W}_d^1\mathbf{z} + \mathbf{b}_d^1)_+ + \mathbf{b}_d^2 \tag{14}$$

where the linear operators ($\mathbf{W}$ matrices) can optionally be constrained (for example for convolution operations $\mathbf{W}$ could be required to be a Toeplitz matrix). Now if we have that the embedded dimension $d$ is equal to the data dimension $d_x$ we will say that linear operators can **express identity on X**

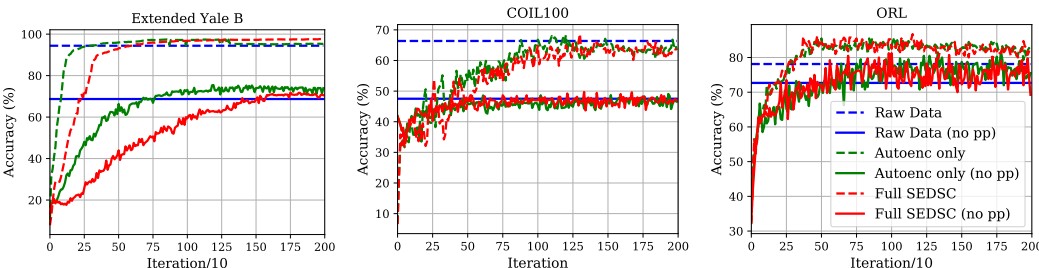

Figure 2: Clustering accuracy results for YaleB (38 faces), COIL100, and ORL datasets with **(Dashed Lines)** and without **(Solid Lines)** the post-processing step on **C** matrix proposed in Ji et al. (2017). **(Raw Data)** Clustering on the raw data. **(Autoenc only)** Clustering features from an autoencoder trained without the $F(\mathbf{Z}, \mathbf{C})$ term. **(Full SEDSC)** The full model in (3).

if there exists parameters $(\mathbf{W}^2, \mathbf{W}^1)$ such that $\mathbf{W}^2\mathbf{W}^1\mathbf{X} = \mathbf{X}$. Note that if the architectures in (14) are fully-connected this implies that for a general $\mathbf{X}$ the number of hidden units is greater than or equal to $d = d_x$ (which can be even smaller if $\mathbf{X}$ is low-rank), while if the linear operators are convolutions, then we only need one convolutional channel (with the kernel being the delta function). Within this setting we have the following result:

**Proposition 2.** *Consider encoder and decoder networks with the form given in* (14). *Then, given any dataset* $\mathbf{X} \in \mathbb{R}^{d_x \times N}$ *where the linear operators in both the encoder/decoder can express identity on* $\mathbf{X}$ *and any* $\tau > 0$ *there exist network parameters* $(\mathcal{W}_e, \mathcal{W}_d)$ *which satisfy the following:*

1. *Embedded points are arbitrarily close:* $\|\Phi_E(\mathbf{X}_i, \mathcal{W}_e) - \Phi_E(\mathbf{X}_j, \mathcal{W}_e)\| \leq \epsilon \; \forall (i, j)$ *and* $\forall \epsilon > 0$.
2. *Embedded points have norm arbitrarily close to* $\tau$: $|\|\Phi_E(\mathbf{X}_i, \mathcal{W}_e)\|_F - \tau| \leq \epsilon \; \forall i$ *and* $\forall \epsilon > 0$.
3. *Embedded points can be decoded exactly:* $\Phi_D(\Phi_E(\mathbf{X}_i, \mathcal{W}_e), \mathcal{W}_d) = \mathbf{X}_i, \; \forall i$.

From the above simple example, we can see that even with very simple network architectures (i.e., not necessarily highly expressive) it is still possible to have solutions which are arbitrarily close to the global optimum described in Theorem 4, in the sense that the points can be made to be arbitrarily close to each other in the embedded space with norm arbitrarily close to $\tau$ (for any arbitrary choice of $\tau$), while still having a perfect reconstruction of $\mathbf{X}$.

## 3 EXPERIMENTS

Here, we present experiments on both real and synthetic data that verify our theoretical predictions experimentally. We first evaluate the Autoencoder Regularization form given in (3) by repeating all of the experiments from Ji et al. (2017). In the Supplementary Material we first show that the optimization problem never reaches a stationary point due to the pathology described by Prop 1 (see Figure 4 in Supplementary Material), and below we show that the improvement in performance reported in Ji et al. (2017) is largely attributable to an ad-hoc post-processing step. Then, we present experiments on a simple synthetic dataset to illustrate our theoretical results.

**Repeating the Experiments of Ji et al. (2017).** First we use the code provided by the authors of Ji et al. (2017) to repeat all of their original clustering experiments on the Extended Yale-B (38 faces), ORL, and COIL100 datasets. As baseline methods, we perform subspace clustering on the raw data as well as subspace clustering on embedded features obtained by training the autoencoder network without the $F(\mathbf{Z}, \mathbf{C})$ term (i.e., $\gamma = 0$ and $\mathbf{C}$ fixed at $\mathbf{I}$ in (3)). See Supplementary Material for further details.

In addition to proposing the model in (3), the authors of Ji et al. (2017) also implement a somewhat arbitrary post-processing of the **C** matrix recovered from the SEDSC model before the final spectral

Table 1: Clustering accuracy shown in Fig 2. To be consistent with Ji et al. (2017), we report the results at 1000/120/700 iterations for Yale B / COIL100 / ORL, respectively.

| | With post-processing | | | Without post-processing | | |
|---|---|---|---|---|---|---|
| | Raw Data | Autoenc only | Full SEDSC | Raw Data | Autoenc only | Full SEDSC |
| YaleB | 94.40% | **97.12**% | 96.79% | 68.71% | **71.96**% | 59.09% |
| COIL100 | 66.47% | **68.26**% | 64.96% | **47.51**% | 44.84% | 45.67% |
| ORL | 78.12% | 83.43% | **84.10**% | 72.68% | **73.73**% | 73.50% |

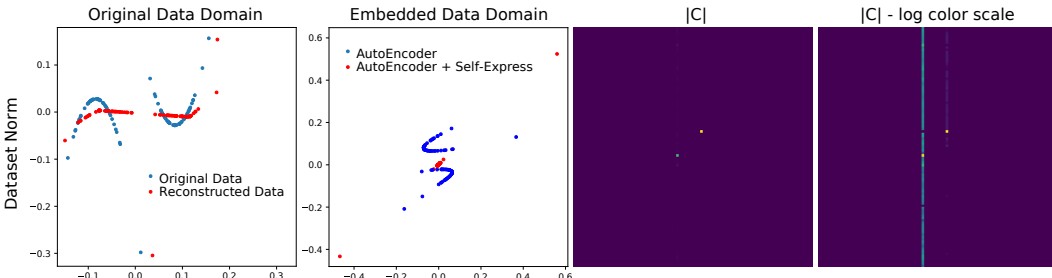

Figure 3: Results for synthetic data using the dataset normalization scheme. **(Left)** Original data points (Blue) and the data points at the output of the autoencoder when the full model (3) is used (Red). **(Center Left)** Data representation in the embedded domain when just the autoencoder is trained without the $F(\mathbf{Z}, \mathbf{C})$ term (Blue) and the full SEDSC model is used (Red). **(Center Right)** The absolute value of the recovered $\mathbf{C}$ encoding matrix when trained with the full model. **(Right)** Same plot as the previous column but with a logarithmic color scale to visualize small entries.

clustering, which involves 1) entrywise hard thresholding, 2) applying the shape interaction matrix method Costeira & Kanade (1995) to $\mathbf{C}$, and 3) raising $\mathbf{C}$ to a power, entry-wise. Likewise, many subsequent works on SEDSC follow Ji et al. (2017) and employ very similar post-processing steps on $\mathbf{C}$. As shown in Figure 2 and Table 1 there is little observed benefit for using SEDSC, as it achieves comparable (or worse) performance than baseline methods in almost all settings when the post-processing of $\mathbf{C}$ is applied consistently (or not) across all methods.

**Synthetic Data Experiments.** Finally, to illustrate our theoretical predictions we construct a simple synthetic dataset which consists of 100 points drawn from the union of two parabolas in $\mathbb{R}^2$, where the space of the embedding is also $\mathbb{R}^2$. We then train the model given in (3) with $\theta(\mathbf{C}) = \theta_{SSC}(\mathbf{C}) = \|\mathbf{C}\|_1 + \delta(\text{diag}(\mathbf{C}) = \mathbf{0})$, the encoder/decoder networks being simple single-hidden-layer fully-connected networks with 100 hidden units, and ReLU activations on the hidden units. Figure 3 shows the solution obtained when we directly add a normalization operator to the encoder network which normalizes the output of the encoder to have unit Frobenius norm over the entire dataset (Dataset Normalization). Additional experiments for other normalization schemes and a description of the details of our experiments can be found in the Supplementary Material.

From Figure 3 one can see that our theoretical predictions are largely confirmed experimentally. Namely, one sees that when the full SEDSC model is trained the embedded representation largely as predicted by Theorem 2, with almost all of the embedded points (Left Center - Red points) being close to the origin with the exception of two points, which are co-linear with each other. Likewise, the $\mathbf{C}$ matrix is dominated by two non-zero entries with the remaining non-zero entries only appearing on the log-scale color scale. We note that as this is a very simple dataset (i.e., two parabolas without any added noise) one would expect most reasonable manifold clustering/learning algorithms to succeed; however, due to the deficiencies of the SEDSC model we have shown in our analysis a trivial solution results.

## 4 CONCLUSIONS

We have presented a theoretical and experimental analysis of the Self-Expressive Deep Subspace Clustering (SEDSC) model. We have shown that in many cases the SEDSC model is ill-posed and results in trivial data geometries in the embedded space. Further, our attempts to replicate previously reported experiments lead us to conclude that much of the claimed benefit of SEDSC is attributable to other factors such as post-processing of the recovered encoding matrix, $\mathbf{C}$, and not the SEDSC model itself. Overall, we conclude that considerably more attention needs to be given to the issues we have raised in this paper in terms of both how models for this problem are designed and how they are evaluated to ensure that one arrives at meaningful solutions and can clearly demonstrate the performance of the resulting model without other confounding factors.

**Acknowledgments.** The authors thank Zhihui Zhu and Benjamin Béjar Haro for helpful discussions in the early stages of this work. This work was partially supported by the Northrop Grumman Mission Systems Research in Applications for Learning Machines (REALM) initiative, NSF Grants 1704458, 2031985 and 1934979, and the Tsinghua-Berkeley Shenzhen Institute Research Fund.

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

# 5 PROOFS OF RESULTS IN MAIN PAPER

Here we present the proofs of our various results that we give in the main paper.

## 5.1 PROOF OF PROPOSITION 1

**Proposition 1.** *Consider the objective in* (3) *and suppose* $\theta(\mathbf{C})$ *in* (1) *is a function which achieves its minimum at* $\mathbf{C} = \mathbf{0}$ *and satisfies* $\theta(\mu\mathbf{C}) < \theta(\mathbf{C})$, $\forall \mu \in (0,1)$. *If for any choice of* $(\mathcal{W}_d, \mathcal{W}_e)$ *and* $\tau_1, \tau_2 \in (0,1)$ *there exists* $(\hat{\mathcal{W}}_d, \hat{\mathcal{W}}_e)$ *such that* $\Phi_E(\mathbf{X}, \hat{\mathcal{W}}_e) = \tau_1 \Phi_E(\mathbf{X}, \mathcal{W}_e)$ *and* $\Phi_D(\tau_2\mathbf{Z}, \hat{\mathcal{W}}_d) = \Phi_D(\mathbf{Z}, \mathcal{W}_d)$ *then the* $F$ *term in* (3) *can be made arbitrarily small without changing the value of the loss function* $\ell$. *Further, if the final layer of the encoding network is positively-homogeneous with degree* $p \neq 0$, *such a* $(\hat{\mathcal{W}}_d, \hat{\mathcal{W}}_e)$ *will always exist simply by scaling the network weights of the linear (or affine) operator parameters.*

*Proof.* Let $(\mathbf{C}, \mathcal{W}_e, \mathcal{W}_d)$ be an arbitrary triplet. For any choice of $\tau, \mu \in (0,1)$, note the statement conditions require that there exists $(\hat{\mathbf{C}}, \hat{\mathcal{W}}_e, \hat{\mathcal{W}}_d)$ which satisfy

$$\hat{\mathbf{C}} = \mu\mathbf{C}$$
$$\Phi_E(\mathbf{X}, \hat{\mathcal{W}}_e) = \tau\Phi_E(\mathbf{X}, \mathcal{W}_e) \tag{15}$$
$$\Phi_D(\mu\tau\mathbf{Z}, \hat{\mathcal{W}}_d) = \Phi_D(\mathbf{Z}, \mathcal{W}_d).$$

Then, the function $\ell$ evaluated at $(\hat{\mathbf{C}}, \hat{\mathcal{W}}_e, \hat{\mathcal{W}}_d)$ is given by

$$\ell(\mathbf{X}, \Phi_D(\Phi_E(\mathbf{X}, \hat{\mathcal{W}}_e)\mu\mathbf{C}, \hat{\mathcal{W}}_d)) = \ell(\mathbf{X}, \Phi_D(\Phi_E(\mathbf{X}, \mathcal{W}_e)\mathbf{C}, \mathcal{W}_d)), \tag{16}$$

which is equal to $\ell$ evaluated at $(\mathbf{C}, \mathcal{W}_e, \mathcal{W}_d)$. Moreover, the function $F$ evaluated at $(\hat{\mathbf{C}}, \hat{\mathcal{W}}_e, \hat{\mathcal{W}}_d)$ is given by

$$\frac{1}{2}\|\Phi_E(\mathbf{X}, \hat{\mathcal{W}}_e) - \Phi_E(\mathbf{X}, \hat{\mathcal{W}}_e)\mu\mathbf{C}\|_F^2 + \lambda\theta(\mu\mathbf{C})$$
$$= \tau^2 \frac{1}{2}\|\Phi_E(\mathbf{X}, \mathcal{W}_e) - \Phi_E(\mathbf{X}, \mathcal{W}_e)\mu\mathbf{C}\|_F^2 + \lambda\theta(\mu\mathbf{C}). \tag{17}$$

Note that the above equation can be made arbitrarily small for choice of $\tau, \mu$ sufficiently small, completing the first statement of the result.

To see the second part of the claim, first note that the condition on the decoder – that for all $\tau_2 \in (0,1)$ there exists $\hat{\mathcal{W}}_d$ such that $\Phi_D(\tau_2\mathbf{Z}, \hat{\mathcal{W}}_d) = \Phi_D(\mathbf{Z}, \mathcal{W}_d)$ – is trivially satisfied by all neural networks by simply scaling the input weights in the first layer of the decoder network by $\tau_2^{-1}$. As a result, we are left to show that there will always exist a set of encoder weights which satisfies the conditions of the statement. To see this, w.l.o.g. let $\Phi_e(\mathbf{X}, \mathcal{W}_e)$ take the general form

$$\Phi_e(\mathbf{X}, \mathcal{W}_e) = \psi(\mathcal{A}(h(\mathbf{X}, \bar{\mathcal{W}}_e), \mathbf{A}, \mathbf{b})) \tag{18}$$

where $\mathcal{A}(\cdot, \mathbf{A}, \mathbf{b})$ is an arbitrary linear (or affine) operator parameterized by linear parameters $\mathbf{A}$ and bias terms (for affine operators) $\mathbf{b}$; $h(\mathbf{X}, \bar{\mathcal{W}}_e)$ is an arbitrary function parameterized by $\bar{\mathcal{W}}_e$ (note that $\mathcal{W}_e = \{\mathbf{A}, \mathbf{b}, \bar{\mathcal{W}}_e\}$); and $\psi$ is an arbitrary positively homogeneous function with degree $p \neq 0$. From this note that for any $\alpha > 0$ we have the following:

$$\psi(\mathcal{A}(h(\mathbf{X}, \bar{\mathcal{W}}_e), \alpha\mathbf{A}, \alpha\mathbf{b})) = \psi(\alpha\mathcal{A}(h(\mathbf{X}, \bar{\mathcal{W}}_e), \mathbf{A}, \mathbf{b})) = \alpha^p\psi(\mathcal{A}(h(\mathbf{X}, \bar{\mathcal{W}}_e), \mathbf{A}, \mathbf{b})). \tag{19}$$

where the first equality is due to basic properties of linear (or affine) operators and the second equality is due to positive homogeneity. As a result, for any choice of $\tau_1 \in (0,1)$ we can choose a scaling $\alpha = \tau_1^{1/p}$ to achieve that for parameters $\hat{\mathcal{W}}_e = (\tau_1^{1/p}\mathbf{A}, \tau_1^{1/p}\mathbf{b}, \bar{\mathcal{W}}_e)$ we have $\Phi_e(\mathbf{X}, \hat{\mathcal{W}}_e) = \tau_1 \Phi_e(\mathbf{X}, \mathcal{W}_e)$, completing the result. $\square$

## 5.2 PROOF OF THEOREM 1

**Theorem 1.** *Consider the following optimization problems which jointly optimize over* $\mathbf{Z}$ *and* $\mathbf{C}$:

$$\text{(P1)} \quad \min_{\mathbf{C}, \mathbf{Z}} \left\{ F(\mathbf{Z}, \mathbf{C}) \text{ s.t. } \|\mathbf{Z}\|_F^2 \geq \tau \right\} \quad \text{(P2)} \quad \min_{\mathbf{C}, \mathbf{Z}} \left\{ F(\mathbf{Z}, \mathbf{C}) \text{ s.t. } \|\mathbf{Z}^i\|_F^2 \geq \frac{\tau}{d} \ \forall i \right\} \tag{20}$$

*Then optimal values for $\mathbf{C}$ for both* (P1) *and* (P2) *are given by*

$$\mathbf{C}^* \in \arg\min_{\mathbf{C}} \tfrac{1}{2}\sigma_{\min}^2(\mathbf{C}-\mathbf{I})\tau + \lambda\theta(\mathbf{C}). \tag{21}$$

*Moreover, for any optimal $\mathbf{C}^*$, let $r$ be the multiplicity of the smallest singular value of $\mathbf{C}^* - \mathbf{I}$ and let $\mathbf{Q} \in \mathbb{R}^{N \times r}$ be an orthonormal basis for the subspace spanned by the left singular vectors associated with the smallest singular value of $\mathbf{C}^* - \mathbf{I}$. Then we have that optimal values for $\mathbf{Z}$ are given by:*

$$\mathbf{Z}^* \in \{\mathbf{B}\mathbf{Q}^\top \colon \mathbf{B} \in \mathbb{R}^{d\times r} \cap \mathcal{B}\}, \quad \mathcal{B} = \begin{cases} \begin{cases} \{\mathbf{B} : \|\mathbf{B}\|_F^2 = \tau\} & \sigma_{\min}(\mathbf{C}^*-\mathbf{I}) > 0 \\ \{\mathbf{B} : \|\mathbf{B}\|_F^2 \geq \tau\} & \sigma_{\min}(\mathbf{C}^*-\mathbf{I}) = 0 \end{cases} & \text{(P1)} \\ \begin{cases} \{\mathbf{B} : \|\mathbf{B}^i\|_F^2 = \frac{\tau}{d}, \forall i\} & \sigma_{\min}(\mathbf{C}^*-\mathbf{I}) > 0 \\ \{\mathbf{B} : \|\mathbf{B}^i\|_F^2 \geq \frac{\tau}{d}, \forall i\} & \sigma_{\min}(\mathbf{C}^*-\mathbf{I}) = 0 \end{cases} & \text{(P2)} \end{cases} \tag{22}$$

*Proof.* The objective $F(\mathbf{Z}, \mathbf{C})$ can be reformulated as

$$F(\mathbf{Z},\mathbf{C}) = \frac{1}{2}\mathrm{tr}(\mathbf{Z}(\mathbf{C}-\mathbf{I})(\mathbf{C}-\mathbf{I})^\top\mathbf{Z}^\top) + \lambda\theta(\mathbf{C}) = \frac{1}{2}\sum_{i=1}^{d}\mathbf{Z}^i(\mathbf{C}-\mathbf{I})(\mathbf{C}-\mathbf{I})^\top(\mathbf{Z}^i)^\top + \lambda\theta(\mathbf{C}), \tag{23}$$

where recall $\mathbf{Z}^i$ denotes the $i^{th}$ row of $\mathbf{Z}$. If we add the constraints on $\mathbf{Z}$ we have the following minimization problems over $\mathbf{Z}$ with $\mathbf{C}$ held fixed:

$$\min_{\mathbf{Z}} \frac{1}{2}\sum_{i=1}^{d}\mathbf{Z}^i(\mathbf{C}-\mathbf{I})(\mathbf{C}-\mathbf{I})^\top(\mathbf{Z}^i)^\top \quad \text{s.t. } \mathbf{Z} \in \mathcal{Z} = \begin{cases} \sum_{i=1}^{d}\|\mathbf{Z}^i\|_F^2 \geq \tau & \text{(P1)} \\ \|\mathbf{Z}^i\|_F^2 \geq \frac{\tau}{d} \,\forall i & \text{(P2)} \end{cases}. \tag{24}$$

Note that if we fix the magnitude of the rows, $\|\mathbf{Z}^i\|_F^2 = \mathbf{k}_i$ for any $\mathbf{k} \in \mathbb{R}^d$, $\mathbf{k} \geq 0$, $\sum_{i=1}^{d}\mathbf{k}_i \geq \tau$ for (P1) and $\mathbf{k} \in \mathbb{R}^d$, $\mathbf{k}_i \geq \tau/d$ for (P2) and optimize over the directions of the rows, then the minimum is obtained whenever $\mathbf{Z}^i$ is in the span of the eigenvectors of $(\mathbf{C}-\mathbf{I})(\mathbf{C}-\mathbf{I})^\top$ with smallest eigenvalue, which implies that all the rows of an optimal $\mathbf{Z}$ matrix must lie in the span of $\mathbf{Q}$, where $\mathbf{Q} \in \mathbb{R}^{N \times r}$ is an orthonormal basis for the subspace by the left singular vectors of $\mathbf{C}-\mathbf{I}$ associated with the smallest singular value of $\mathbf{C}-\mathbf{I}$, which has multiplicity $r$.

As a result, we have that optimal values of $\mathbf{Z}$ must take the form $\mathbf{Z} = \mathbf{B}\mathbf{Q}^\top$ for some $\mathbf{B} \in \mathbb{R}^{d\times r}$. Further, we note that the following also holds:

$$F(\mathbf{B}\mathbf{Q}^\top,\mathbf{C}) = \frac{1}{2}\mathrm{tr}(\mathbf{B}\mathbf{Q}^\top(\mathbf{C}-\mathbf{I})(\mathbf{C}-\mathbf{I})^\top\mathbf{Q}\mathbf{B}^\top) + \lambda\theta(\mathbf{C}) =$$
$$= \frac{1}{2}\sigma_{\min}^2(\mathbf{C}-\mathbf{I})\mathrm{tr}(\mathbf{B}\mathbf{B}^\top) + \lambda\theta(\mathbf{C}) = \frac{1}{2}\sigma_{\min}^2(\mathbf{C}-\mathbf{I})\|\mathbf{B}\|_F^2 + \lambda\theta(\mathbf{C}). \tag{25}$$

The constraints for $\mathbf{B}$ are then seen by noting $\mathbf{Z}\mathbf{Z}^\top = \mathbf{B}\mathbf{Q}^\top\mathbf{Q}\mathbf{B}^\top = \mathbf{B}\mathbf{B}^\top$, so $\|\mathbf{Z}\|_F^2 = \mathrm{tr}(\mathbf{Z}\mathbf{Z}^\top) = \mathrm{tr}(\mathbf{B}\mathbf{B}^\top) = \|\mathbf{B}\|_F^2$ and $\|\mathbf{Z}^i\|_F^2 = (\mathbf{Z}\mathbf{Z}^\top)_{i,i} = (\mathbf{B}\mathbf{B}^\top)_{i,i} = \mathbf{B}^i(\mathbf{B}^i)^\top = \|\mathbf{B}^i\|_F^2$, and if $\sigma_{\min}(\mathbf{C}^*-\mathbf{I}) > 0$ then minimizing (25) w.r.t. $\mathbf{B}$ subject to the constraints on $\mathbf{Z}$ gives that $\|\mathbf{B}\|_F^2 = \tau$ is optimal. $\qquad\square$

### 5.3 PROOF OF THEOREM 2

**Theorem 2.** *Optimal solutions to the problems*

$$\text{(P1)} \quad \min_{\mathbf{Z},\mathbf{C}} \|\mathbf{C}\|_1 \quad \text{s.t. } \mathrm{diag}(\mathbf{C}) = \mathbf{0}, \ \mathbf{Z} = \mathbf{Z}\mathbf{C}, \ \|\mathbf{Z}\|_F^2 \geq \tau \tag{26}$$

$$\text{(P2)} \quad \min_{\mathbf{Z},\mathbf{C}} \|\mathbf{C}\|_1 \quad \text{s.t. } \mathrm{diag}(\mathbf{C}) = \mathbf{0}, \ \mathbf{Z} = \mathbf{Z}\mathbf{C}, \ \|\mathbf{Z}^i\|_F^2 \geq \frac{\tau}{d} \ \forall i \tag{27}$$

*are characterized by the set*

$$(\mathbf{Z}^*,\mathbf{C}^*) \in \left\{ \begin{bmatrix} \mathbf{z} & \mathbf{z} & \mathbf{0}_{d\times N-2} \end{bmatrix}\mathbf{P} \right\} \times \left\{ \mathbf{P}^\top \begin{bmatrix} 0 & 1 & 0 & \cdots & 0 \\ 1 & 0 & 0 & \cdots & 0 \\ 0 & 0 & 0 & \cdots & 0 \\ \vdots & \vdots & \vdots & & \ddots \\ 0 & 0 & 0 & & 0 \end{bmatrix}\mathbf{P} \right\}, \tag{28}$$

*where $\mathbf{P} \in \mathbb{R}^{N \times N}$ is an arbitrary signed-permutation matrix and $\mathbf{z} \in \mathbb{R}^d$ is an arbitrary vector which satisfies $\|\mathbf{z}\|_F^2 \geq \tau/2$ for (P1) and $\mathbf{z}_i^2 \geq \tau/(2d), \forall i \in [1,d]$ for (P2).*

*Proof.* As we observed from the proof of Theorem 1, if $\mathbf{C} - \mathbf{I}$ has a left null-space we can choose an optimal $\mathbf{Z}$ to have its rows in that null-space (and this also clearly implies we have $\sigma_{\min}(\mathbf{C} - \mathbf{I}) = 0$). Also note that when $\sigma_{\min}(\mathbf{C} - \mathbf{I}) = 0$ this corresponds to the case where $\mathbf{Z} = \mathbf{Z}\mathbf{C} \iff \mathbf{Z}(\mathbf{I} - \mathbf{C}) = \mathbf{0}$. Further, note that if $\mathbf{q}$ is a non-zero vector in the left null-space of $\mathbf{C} - \mathbf{I}$ we have that $\mathbf{q}^\top(\mathbf{C} - \mathbf{I}) = \mathbf{0} \iff \mathbf{q}^\top \mathbf{C} = \mathbf{q}^\top$, which implies that if we take $\mathbf{q}$ to be all-zero except for its $i^{\text{th}}$ entry, then this would imply that $\mathbf{C}_{i,i}$ must be non-zero, which would violate the $\text{diag}(\mathbf{C}) = \mathbf{0}$ constraint, so any vector $\mathbf{q}$ in a left null-space of $\mathbf{C} - \mathbf{I}$ for a feasible $\mathbf{C}$ matrix must have at least two non-zero entries. As a result, solving (5) with the constraint $\sigma_{\min}(\mathbf{C} - \mathbf{I}) = 0$ is equivalent to the following problem:

$$\min_{\mathbf{C}, \mathbf{q}} \|\mathbf{C}\|_1 \ \text{ s.t. } \ \mathbf{q}^\top(\mathbf{C} - \mathbf{I}) = \mathbf{0}, \ \text{diag}(\mathbf{C}) = \mathbf{0}, \ \|\mathbf{q}\|_0 \geq 2. \tag{29}$$

where $\| \cdot \|_0$ denotes the $\ell_0$ pseudo-norm of a vector defined as the number of non-zero entries in a vector. To first minimize w.r.t. $\mathbf{C}$ with a fixed $\mathbf{q}$, we form the Lagrangian:

$$\min_{\mathbf{C}} \left\{ L(\mathbf{C}, \mathbf{\Lambda}, \mathbf{\Gamma}) = \|\mathbf{C}\|_1 + \langle \mathbf{\Lambda}, (\mathbf{I} - \mathbf{C}^\top)\mathbf{q} \rangle + \langle \text{Diag}(\mathbf{\Gamma}), \mathbf{C} \rangle \right\} = \tag{30}$$

$$\min_{\mathbf{C}} \|\mathbf{C}\|_1 + \langle \mathbf{q}\mathbf{\Lambda}^\top - \text{Diag}(\mathbf{\Gamma}), \mathbf{C} \rangle + \langle \mathbf{\Lambda}, \mathbf{q} \rangle = \langle \mathbf{\Lambda}, \mathbf{q} \rangle - \delta(\|\mathbf{q}\mathbf{\Lambda}^\top - \text{Diag}(\mathbf{\Gamma})\|_\infty \leq 1), \tag{31}$$

where $\mathbf{\Lambda} \in \mathbb{R}^N$ and $\mathbf{\Gamma} \in \mathbb{R}^N$ are vectors of dual variables to enforce the $(\mathbf{I} - \mathbf{C}^\top)\mathbf{q} = \mathbf{0}$ and the $\text{diag}(\mathbf{C}) = \mathbf{0}$ constraints, respectively. This gives the dual problem

$$\max_{\mathbf{\Lambda}, \mathbf{\Gamma}} \left\{ \langle \mathbf{\Lambda}, \mathbf{q} \rangle \ \text{ s.t. } \ \|\mathbf{q}\mathbf{\Lambda}^\top - \text{Diag}(\mathbf{\Gamma})\|_\infty \leq 1 \right\} = \max_{\mathbf{\Lambda}} \left\{ \langle \mathbf{\Lambda}, \mathbf{q} \rangle \ \text{ s.t. } \ |\mathbf{q}_i \mathbf{\Lambda}_j| \leq 1 \ \forall i \neq j \right\}. \tag{32}$$

We note that (32) is separable in the entries of $\mathbf{\Lambda}$, so if we define $\{i_k\}_{k=1}^N$ to be the indexing which sorts the absolute values of the entries of $\mathbf{q}$ in descending order, $|\mathbf{q}_{i_1}| \geq |\mathbf{q}_{i_2}| \geq \cdots \geq |\mathbf{q}_{i_N}|$, one can easily see that an optimal choice of $\mathbf{\Lambda}$ is given by

$$\mathbf{\Lambda}_{i_1}^* = \frac{\text{sgn}(\mathbf{q}_{i_1})}{|\mathbf{q}_{i_2}|}, \quad \mathbf{\Lambda}_{i_k}^* = \frac{\text{sgn}(\mathbf{q}_{i_k})}{|\mathbf{q}_{i_1}|} \ \forall k \in [2, N] \implies \langle \mathbf{\Lambda}^*, \mathbf{q} \rangle = \frac{|\mathbf{q}_{i_1}|}{|\mathbf{q}_{i_2}|} + \frac{1}{|\mathbf{q}_{i_1}|} \sum_{k=2}^N |\mathbf{q}_{i_k}|. \tag{33}$$

If we now minimize the above w.r.t. $\mathbf{q}$, note that the optimal value of the dual objective given by the above equation is invariant w.r.t. scaling the $\mathbf{q}$ vector by any non-zero scalar, so we can w.l.o.g. assume that $|\mathbf{q}_{i_1}| = 1$ and note that this implies that problem (29) is equivalent to the following optimization problem over the magnitudes of $\mathbf{q}$ if we define $p_k = |\mathbf{q}_{i_k}|$:

$$\min_{\{p_k\}_{k=2}^N} \frac{1}{p_2} + p_2 + \sum_{k=3}^N p_k \ \text{ s.t. } \ 1 \geq p_2 \geq p_3 \geq \cdots \geq p_N \geq 0. \tag{34}$$

Now, note that for a non-negative scalar $\alpha \geq 0$ the minimum of $\alpha^{-1} + \alpha$ is achieved at $\alpha = 1$, so one can clearly see that the optimal value for the above problem is achieved at $p_2 = 1$ and $p_k = 0, \ \forall k \in [3, N]$. From this we have that an optimal $\mathbf{q}$ for (29) must have exactly two non-zero entries and the non-zero entries must be equal in absolute value. Further, this also implies that $\|\mathbf{C}^*\|_1 = 2$, and because we must have $\mathbf{q}^\top \mathbf{C}^* = \mathbf{q}^\top$, if we scale $\mathbf{q}$ to have $\pm 1$ for its two non-zero entries, we then have $\|\mathbf{q}^\top \mathbf{C}^*\|_1 = \|\mathbf{q}\|_1 = 2 = \|\mathbf{C}^*\|_1$, so if we let $(i, j)$ index the two non-zero entries of $\mathbf{q}$ we have:

$$2 = \|\mathbf{q}^\top \mathbf{C}^*\|_1 = \|\text{sgn}(\mathbf{q}_i)(\mathbf{C}^*)^i + \text{sgn}(\mathbf{q}_j)(\mathbf{C}^*)^j\|_1 \leq \|(\mathbf{C}^*)^i\|_1 + \|(\mathbf{C}^*)^j\|_1 \leq \|\mathbf{C}^*\|_1 = 2. \tag{35}$$

This implies that all the non-zero entries of $\mathbf{C}^*$ must lie in rows $i$ and $j$, and if there is any overlap in the non-zero support of these rows the signs must match after multiplication by $\text{sgn}(\mathbf{q}_i)$ and $\text{sgn}(\mathbf{q}_j)$. However, since $\mathbf{q}^\top \mathbf{C}$ must equal $\mathbf{q}^\top$ (which is zero everywhere except for entries $i$ and $j$) and the diagonal of $\mathbf{C}^*$ must be zero, the only way this can be achieved is for the two rows to have non-overlapping non-zero support, proving that the only non-zero entries of $\mathbf{C}$ must be $\mathbf{C}_{i,j}$ and $\mathbf{C}_{j,i}$ which take values in $\{-1, 1\}$, depending on the choice of the signs for $\mathbf{q}_i$ and $\mathbf{q}_j$. The result is completed by noting that since we require $\mathbf{Z}^* = \mathbf{Z}^* \mathbf{C}^*$, then $\mathbf{Z}_i^*$ and $\mathbf{Z}_j^*$ must be equal to within a sign-flip depending on the choice of the signs of the $\mathbf{q}$ vector. $\qquad \square$

## 5.4 PROOF OF THEOREM 3

**Theorem 3.** *Optimal solutions to the problems*

$$\text{(P1)} \quad \min_{\mathbf{Z},\mathbf{C}} \|\mathbf{C}\|_{\mathcal{S}_p} \text{ s.t. } \mathbf{Z} = \mathbf{Z}\mathbf{C}, \ \|\mathbf{Z}\|_F^2 \geq \tau \tag{36}$$

$$\text{(P2)} \quad \min_{\mathbf{Z},\mathbf{C}} \|\mathbf{C}\|_{\mathcal{S}_p} \text{ s.t. } \mathbf{Z} = \mathbf{Z}\mathbf{C}, \ \|\mathbf{Z}^i\|_F^2 \geq \tfrac{\tau}{d} \ \forall i, \tag{37}$$

*where $\|\mathbf{C}\|_{\mathcal{S}_p}$ is any Schatten-$p$ norm on $\mathbf{C}$, are characterized by the set*

$$(\mathbf{Z}^*, \mathbf{C}^*) \in \left\{ (\mathbf{z}\mathbf{q}^\top) \times (\mathbf{q}\mathbf{q}^\top) : \mathbf{q} \in \mathbb{R}^N, \|\mathbf{q}\|_F = 1 \right\} \tag{38}$$

*where $\mathbf{z} \in \mathbb{R}^d$ is an arbitrary vector which satisfies $\|\mathbf{z}\|_F^2 \geq \tau$ for (P1) and $\mathbf{z}_i^2 \geq \tfrac{\tau}{d}$, $\forall i$ for (P2).*

*Proof.* To begin, by the same arguments as in Theorem 2 we consider an optimization problem similar to (29) but for $\theta(\mathbf{C}) = \|\mathbf{C}\|_{\mathcal{S}_p}$ being any Schatten-$p$ norm and with the $\|\mathbf{q}\|_0$ constraint replaced by a $\mathbf{q} \neq \mathbf{0}$ constraint:

$$\min_{\mathbf{C},\mathbf{q}} \|\mathbf{C}\|_{\mathcal{S}_p} \text{ s.t. } \mathbf{q}^\top(\mathbf{C} - \mathbf{I}) = \mathbf{0}, \ \mathbf{q} \neq \mathbf{0}. \tag{39}$$

Again forming the Lagrangian for $\mathbf{C}$ with $\mathbf{q}$ fixed we have:

$$\min_{\mathbf{C}} \left\{ L(\mathbf{C}, \Lambda) = \|\mathbf{C}\|_{\mathcal{S}_p} + \langle \Lambda, (\mathbf{I} - \mathbf{C}^\top)\mathbf{q} \rangle \right\} = \tag{40}$$

$$\min_{\mathbf{C}} \|\mathbf{C}\|_{\mathcal{S}_p} - \langle \mathbf{q}\Lambda^\top, \mathbf{C} \rangle + \langle \Lambda, \mathbf{q} \rangle = \langle \Lambda, \mathbf{q} \rangle - \delta(\|\mathbf{q}\Lambda^\top\|_{\mathcal{S}_p}^\circ \leq 1) \tag{41}$$

which implies the dual problem is:

$$\max_{\Lambda} \langle \Lambda, \mathbf{q} \rangle \text{ s.t. } \|\mathbf{q}\Lambda^\top\|_{\mathcal{S}_p}^\circ \leq 1 \tag{42}$$

where $\|\cdot\|_{\mathcal{S}_p}^\circ$ denotes the dual norm. Note that for any Schatten-$p$ norm, the dual norm is again a Schatten-$p$ norm, but since we only evaluate the norm on rank-1 matrices this is equal to the Frobenius norm for all values of $p$. As a result we have for all choices of Schatten-$p$ norm that the dual problem is equivalent to:

$$\max_{\Lambda} \left\{ \langle \Lambda, \mathbf{q} \rangle \text{ s.t. } \|\mathbf{q}\Lambda^\top\|_F \leq 1 \right\} = \max_{\Lambda} \left\{ \langle \Lambda, \mathbf{q} \rangle \text{ s.t. } \|\mathbf{q}\|_F \|\Lambda\|_F \leq 1 \right\} \tag{43}$$

From the above, one can easily see that the optimal choice for $\Lambda$ is given as $\Lambda^* = \frac{\mathbf{q}}{\|\mathbf{q}\|_F^2}$ and the optimal objective value is 1. Further note that from primal optimality in (41) we must have that $\mathbf{q}(\Lambda^*)^\top \in \partial\|\mathbf{C}^*\|_{\mathcal{S}_p}$ which implies that $\langle \mathbf{C}^*, \mathbf{q}(\Lambda^*)^\top \rangle = \|\mathbf{C}^*\|_{\mathcal{S}_p} \|\mathbf{q}(\Lambda^*)^\top\|_{\mathcal{S}_p}^\circ = \|\mathbf{C}^*\|_{\mathcal{S}_p} = 1$. As a result, we have that $\mathbf{C}^* = \mathbf{q}(\Lambda^*)^\top$ by the Cauchy-Schwarz inequality and the fact that $\|\mathbf{q}(\Lambda^*)^\top\|_F = 1$. Thus since $\mathbf{C}^*$ is a rank-1 matrix then $\mathbf{C}^* - \mathbf{I}$ can only have one singular value equal to 0, so all the rows of $\mathbf{Z}^*$ must be equal to a scaling of $\mathbf{q}$. Given this structure for $\mathbf{Z}^*$ the result then follows, where we also recall that $\|\mathbf{C}^*\|_{\mathcal{S}_p} = 1$, which implies the constraints on $\mathbf{q}$ in the statement. □

## 5.5 PROOF OF THEOREM 4

We now present the proof for Theorem 4. Before proving the main result we first prove a simple Lemma which will be helpful.

**Lemma 1.** *For a given matrix $\mathbf{Z} \in \mathbb{R}^{d \times N}$ and vector $\mathbf{z} \in \mathbb{R}^d$ such that $\|\mathbf{Z}_i\|_F = \tau \ \forall i$ and $\|\mathbf{z}\|_F = \tau$, let $k \in [0, N]$ be the number of columns in $\mathbf{Z}$ which are equal to $\mathbf{z}$ to within a sign-flip (i.e., $\mathbf{Z}_i = \pm\mathbf{z}$). Then, if $k \geq 1$ the following holds:*

$$\min_{\mathbf{c}} \left\{ \|\mathbf{c}\|_1 \text{ s.t. } \mathbf{z} = \mathbf{Z}\mathbf{c} \right\} = 1 \tag{44}$$

*and $\mathbf{c}_i^* \neq 0 \implies \mathbf{Z}_i = \pm\mathbf{z}$.*

*Further, if $k = 0$ we also have*

$$\min_{\mathbf{c}} \left\{ \|\mathbf{c}\|_1 \text{ s.t. } \mathbf{z} = \mathbf{Z}\mathbf{c} \right\} > 1 \tag{45}$$

*(where we use the convention that the objective takes value $\infty$ if $\mathbf{z} = \mathbf{Z}\mathbf{c}$ has no feasible solution.)*

*Proof.* Without loss of generality, assume the columns of $\mathbf{Z}$ are permuted so that $\mathbf{Z}$ has the form:

$$\mathbf{Z} = \begin{bmatrix} \mathbf{z}\mathbf{s}^\top, & \bar{\mathbf{Z}} \end{bmatrix} \tag{46}$$

where $\mathbf{s} \in \{-1, 1\}^k$ is a vector with $k \in [0, N]$ elements each taking value $-1$ or $1$, and $\bar{\mathbf{Z}} \in \mathbb{R}^{d \times (N-k)}$ contains all the columns of $\mathbf{Z}$ which are not equal to $\pm \mathbf{z}$.

First we consider the $k \geq 1$ case and note that the Lagrangian of (44) is given as:

$$L(\mathbf{c}, \Lambda) = \|\mathbf{c}\|_1 + \langle \Lambda, \mathbf{z} - \mathbf{Z}\mathbf{c} \rangle \tag{47}$$

Now minimizing $L$ w.r.t. $\mathbf{c}$ gives

$$\min_{\mathbf{c}} \|\mathbf{c}\|_1 - \langle \mathbf{Z}^\top \Lambda, \mathbf{c} \rangle = -\delta(\|\mathbf{Z}^\top \Lambda\|_\infty \leq 1) \tag{48}$$

which gives that the dual problem to (44) (with $k \geq 1$) is given by

$$\max_{\Lambda} \langle \Lambda, \mathbf{z} \rangle \text{ s.t. } \|\Lambda^\top \mathbf{Z}\|_\infty \leq 1 \iff \max_{\Lambda} \langle \Lambda, \mathbf{z} \rangle \text{ s.t. } |\langle \Lambda, \mathbf{Z}_i \rangle| \leq 1, \ \forall i \in [1, N] \iff$$
$$\max_{\Lambda} \langle \Lambda, \mathbf{z} \rangle \text{ s.t. } |\langle \Lambda, \mathbf{z} \rangle| \leq 1, \ |\langle \Lambda, \bar{\mathbf{Z}}_i \rangle| \leq 1, \ \forall i \in [1, N - k] \tag{49}$$

where the final equivalence is due to the special structure of $\mathbf{Z}$ in (46). Clearly from (49) and the fact that $\|\mathbf{z}\|_F = \|\bar{\mathbf{Z}}_i\|_F = \tau$, $\forall i$ it is easily seen that an optimal $\Lambda$ is any vector such that $\langle \Lambda^*, \mathbf{z} \rangle = 1$, so as a result we have that the optimal solution to the problem in (44) has objective value 1. Further, note that when $k \geq 1$, then due to the triangle inequality and the fact that all the vectors in $\mathbf{Z}$ have equal norm we can only achieve $\mathbf{z} = \mathbf{Z}\mathbf{c}^*$ with $\|\mathbf{c}^*\|_1 = 1$ if all the non-zero entries of $\mathbf{c}$ are in the first $k$ entries and the sign of any non-zero element of $\mathbf{c}^*$ must satisfy $sgn(\mathbf{c}_i^*) = \mathbf{s}_i$, $i \in [1, k]$.

To see that (45) is true, first note that an optimal solution to (49) with $k \geq 1$ is to choose $\Lambda^* = \mathbf{z}\tau^{-2}$ and that because $\bar{\mathbf{Z}}_i \neq \pm \mathbf{z}$, $\forall i$ we have $|\langle \bar{\mathbf{Z}}_i, \Lambda^* \rangle| = |\langle \bar{\mathbf{Z}}_i, \mathbf{z}\tau^{-2} \rangle| < \|\bar{\mathbf{Z}}_i\|_F \|\mathbf{z}\tau^{-2}\|_F = 1$. Further, note that the problem in (45) (with $k = 0$) will have an equivalent dual problem to (49), with the $|\langle \Lambda, \mathbf{z} \rangle| \leq 1$ constraint removed, which shows the inequality, as we can always take $\Lambda = \alpha \mathbf{z}\tau^{-2}$ for some $\alpha > 1$ and remain dual feasible, giving a dual value (and hence optimal objective value) for (45) strictly greater than 1. □

With this result we are now ready to prove Theorem 4.

**Theorem 4.** *Optimal solutions to the problem*

$$\min_{\mathbf{Z}, \mathbf{C}} \|\mathbf{C}\|_1 \text{ s.t. } \text{diag}(\mathbf{C}) = \mathbf{0}, \ \mathbf{Z} = \mathbf{Z}\mathbf{C}, \ \|\mathbf{Z}_i\|_F^2 = \tau \ \forall i \tag{50}$$

*must have the property that for any column in $\mathbf{Z}^*$, $\mathbf{Z}_i^*$, there exists another column, $\mathbf{Z}_j^*$ ($i \neq j$), such that $\mathbf{Z}_i^* = s_{i,j}\mathbf{Z}_j^*$ where $s_{i,j} \in \{-1, 1\}$. Further, $\|\mathbf{C}_i^*\|_1 = 1 \ \forall i$ and $\mathbf{C}_{i,j} \neq 0 \implies \mathbf{Z}_i^* = \pm \mathbf{Z}_j^*$.*

*Proof.* First, note that any $\mathbf{Z}^*$ which satisfies the conditions of the Theorem achieves optimal objective value with $\|\mathbf{C}_i^*\|_1 = 1$, $\forall i$ and $\mathbf{C}_{i,j} \neq 0 \implies \mathbf{Z}_i^* = \pm \mathbf{Z}_j^*$ directly from Lemma 1 since when we are finding an optimal $\mathbf{C}_i$ encoding for column $\mathbf{Z}_i^*$ there must exist another column in $\mathbf{Z}^*$ which is equal to $\mathbf{Z}_i^*$ to within a sign-flip ($k \geq 1$ in Lemma 1).

To show that this is optimal, we will proceed by contradiction and assume we have a feasible pair of matrices $(\tilde{\mathbf{Z}}, \tilde{\mathbf{C}})$ which does not satisfy the conditions of the Theorem but $\|\tilde{\mathbf{C}}\|_1 \leq N = \|\mathbf{C}^*\|_1$. Note that because $\tilde{\mathbf{Z}}$ does not satisfy the conditions of the Theorem this implies that at least one column of $\tilde{\mathbf{Z}}$ must be distinct (i.e., $\exists i : \tilde{\mathbf{Z}}_i \neq \pm \tilde{\mathbf{Z}}_j$, $\forall j \neq i$). As a result, for any column $\tilde{\mathbf{Z}}_i$ which is distinct we must have $\|\tilde{\mathbf{C}}_i\|_1 > 1$ from Lemma 1 ($k = 0$ case). If we let $\mathcal{I}$ denote the set of indices of the distinct columns in $\tilde{\mathbf{Z}}$ and $\mathcal{I}^\circ$ the compliment of $\mathcal{I}$ we then have

$$\|\tilde{\mathbf{C}}_i\|_1 = \sum_{i \in \mathcal{I}} \|\tilde{\mathbf{C}}_i\|_1 + \sum_{j \in \mathcal{I}^\circ} \|\tilde{\mathbf{C}}_j\|_1 \tag{51}$$

$$= \sum_{i \in \mathcal{I}} \|\tilde{\mathbf{C}}_i\|_1 + |\mathcal{I}^\circ| \tag{52}$$

$$> |\mathcal{I}| + |\mathcal{I}^\circ| = N \tag{53}$$

where the first equality comes from noting that for any $\tilde{\mathbf{Z}}_j$, $j \in \mathcal{I}^\circ$ corresponds to the $k \geq 1$ situation in Lemma 1 and the inequality comes from noting that any $\tilde{\mathbf{Z}}_i$, $i \in \mathcal{I}$ corresponds to the $k = 0$ situation in Lemma 1 and the fact that $|\mathcal{I}| \geq 1$. We thus have the contradiction and the result is completed. □

## 5.6 Proof of Proposition 2

**Proposition 2.** *Consider encoder and decoder networks with the form given in* (14). *Then, given any dataset* $\mathbf{X} \in \mathbb{R}^{d_x \times N}$ *where the linear operators in both the encoder/decoder can express identity on* $\mathbf{X}$ *and any* $\tau > 0$ *there exist network parameters* $(\mathcal{W}_e, \mathcal{W}_d)$ *which satisfy the following:*

1. *Embedded points are arbitrarily close:* $\|\Phi_E(\mathbf{X}_i, \mathcal{W}_e) - \Phi_E(\mathbf{X}_j, \mathcal{W}_e)\| \leq \epsilon \, \forall (i, j)$ *and* $\forall \epsilon > 0$.
2. *Embedded points have norm arbitrarily close to* $\tau$: $|\|\Phi_E(\mathbf{X}_i, \mathcal{W}_e)\|_F - \tau| \leq \epsilon \, \forall i$ *and* $\forall \epsilon > 0$.
3. *Embedded points can be decoded exactly:* $\Phi_D(\Phi_E(\mathbf{X}_i, \mathcal{W}_e), \mathcal{W}_d) = \mathbf{X}_i, \, \forall i$.

*Proof.* To begin, let $(\tilde{\mathbf{W}}_e^1, \tilde{\mathbf{W}}_e^2)$ and $(\tilde{\mathbf{W}}_d^1, \tilde{\mathbf{W}}_d^2)$ be choices of linear operator parameters such that $\tilde{\mathbf{W}}_e^2 \tilde{\mathbf{W}}_e^1 \mathbf{X} = \tilde{\mathbf{W}}_d^2 \tilde{\mathbf{W}}_d^1 \mathbf{X} = \mathbf{X}$ which always must exist since the operators can express identity on $\mathbf{X}$. Now, for an arbitrary $\alpha > 0$ let $\tilde{\mathbf{b}}_e^1$ be any vector such that $\alpha \tilde{\mathbf{W}}_e^1 \mathbf{X}_i + \mathbf{b}_e^1$ is non-negative for all $i$ (note that this is always possible by taking $\tilde{\mathbf{b}}_e^1$ to be a sufficiently large non-negative vector). Note that now if we choose $\tilde{\mathbf{b}}_e^2 = \mathbf{0}$ we then have $\forall i$ and all $\beta > 0$:

$$(\beta \tilde{\mathbf{W}}_e^2)(\alpha \tilde{\mathbf{W}}_e^1 \mathbf{X}_i + \tilde{\mathbf{b}}_e^1)_+ + \tilde{\mathbf{b}}_e^2 = (\beta \tilde{\mathbf{W}}_e^2)(\alpha \tilde{\mathbf{W}}^1 \mathbf{X}_i + \tilde{\mathbf{b}}_e^1) = \alpha \beta \mathbf{X}_i + \beta \tilde{\mathbf{W}}_e^2 \tilde{\mathbf{b}}_e^1 \qquad (54)$$

where the ReLU function becomes an identity operator due to the fact that we have all non-negative entries. Likewise, we can choose $\tilde{\mathbf{b}}_d^1$ to be an arbitrary vector such that $(\beta^{-1} \tilde{\mathbf{W}}_d^1)(\alpha \beta \mathbf{X}_i + \beta \mathbf{W}_e^2 \tilde{\mathbf{b}}_e^1) + \tilde{\mathbf{b}}_d^1$ is non-negative for all $\mathbf{X}_i$, so if we choose $\tilde{\mathbf{b}}_d^2 = -(\alpha^{-1} \tilde{\mathbf{W}}_d^2)[\tilde{\mathbf{W}}_d^1 \tilde{\mathbf{W}}_e^2 \tilde{\mathbf{b}}_e^1 + \tilde{\mathbf{b}}_d^1]$ we then have:

$$\begin{aligned}
&(\alpha^{-1} \tilde{\mathbf{W}}_d^2)[(\beta^{-1} \tilde{\mathbf{W}}_d^1)(\alpha \beta \mathbf{X}_i + \beta \tilde{\mathbf{W}}_e^2 \tilde{\mathbf{b}}_e^1) + \tilde{\mathbf{b}}_d^1]_+ + \tilde{\mathbf{b}}_d^2 \\
=&(\alpha^{-1} \tilde{\mathbf{W}}_d^2)[(\beta^{-1} \tilde{\mathbf{W}}_d^1)(\alpha \beta \mathbf{X}_i + \beta \tilde{\mathbf{W}}_e^2 \tilde{\mathbf{b}}_e^1) + \tilde{\mathbf{b}}_d^1] + \tilde{\mathbf{b}}_d^2 \\
=&\tilde{\mathbf{W}}_d^2 \tilde{\mathbf{W}}_d^1 \mathbf{X}_i + (\alpha^{-1} \tilde{\mathbf{W}}_d^2)[\tilde{\mathbf{W}}_d^1 \tilde{\mathbf{W}}_e^2 \tilde{\mathbf{b}}_e^1 + \tilde{\mathbf{b}}_d^1] + \tilde{\mathbf{b}}_d^2 \\
=&\mathbf{X}_i
\end{aligned} \qquad (55)$$

So as a result we have constructed a set of encoder/decoder weights which satisfies the third condition of the statement. Further, the embedded points in this construction are of the form

$$\mathbf{Z}_i = \alpha \beta \mathbf{X}_i + \beta \tilde{\mathbf{W}}_e^2 \tilde{\mathbf{b}}_e^1 \qquad (56)$$

so since we can form such a construction for an arbitrary $\alpha > 0$ and $\beta > 0$ we can choose $\alpha \to 0$ arbitrarily small and $\beta = \tau \|\tilde{\mathbf{W}}_e^2 \tilde{\mathbf{b}}_e^1\|_F^{-1}$ to give that all the embedded points $\mathbf{Z}_i$ are arbitrarily close to the point $\tau \tilde{\mathbf{W}}_e^2 \tilde{\mathbf{b}}_e^1 \|\tilde{\mathbf{W}}_e^2 \tilde{\mathbf{b}}_e^1\|_F^{-1}$, which completes the result. $\square$

## 6 Additional Results and Details

Here we give a few additional results which expand on results given in the main paper along with extra details regarding our experiments.

### 6.1 Experiments with Real Data

In addition to the results we show in the main paper, we also present additional experimental results on real data. In particular Figure 4 (Left) shows the magnitude of the embedded representation, $\mathbf{Z}$, using the original code from Ji et al. (2017) to solve model (3) using the YaleB dataset (38 faces). Note that the optimization never reaches a stationary point with the magnitude of the embedded representation continually decreasing (as predicted by Proposition 1). Additionally, if one looks at the singular values (normalized by the largest singular value) for the embedding of data points from one class (Right), then training the autoencoder without the $F(\mathbf{Z}, \mathbf{C})$ term results in a geometry that is closer to a linear subspace. Further, the raw data is actually closer to a linear subspace than after training the full SEDSC model (comparing Red and Blue curves). Interestingly, the fact that the autoencoder features and raw data is closer to a linear subspace than SEDSC is also consistent with the clustering performance we show in Table 1, where for the setting without the post-processing the autoencoder-only features achieve the best clustering results, followed by the raw data, followed by SEDSC.

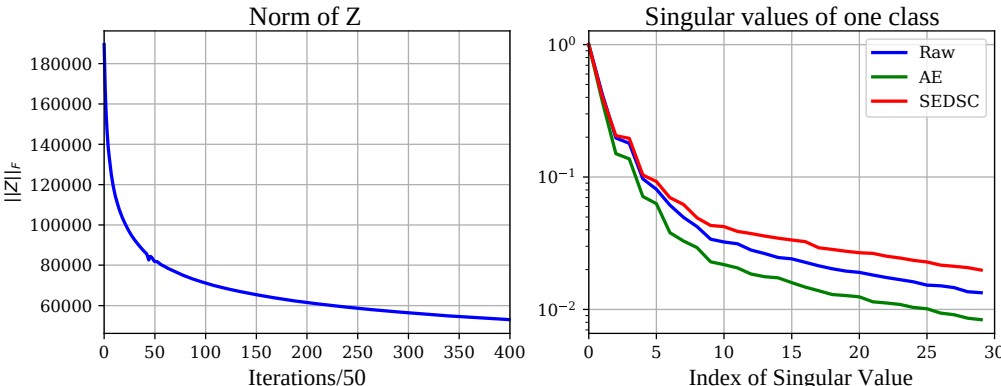

Figure 4: Experiments on Extended Yale B dataset. **(Left)** The norm of the embedded representation $\mathbf{Z}$ as training proceeds. **(Right)** The singular values of the embedded representation of points from one class, normalized by the largest singular value. (Raw) The singular values of the raw data. (AE) The singular values of the embedded representation from an autoencoder trained without the $F(\mathbf{Z}, \mathbf{C})$ term. (SEDSC) The singular values of the embedded representation from the full SEDSC model (3).

**Details of experiments with real data.** We use the code provided by the authors of Ji et al. (2017)[3]. The code implements the model in (3) with $\theta(\mathbf{C}) = \frac{1}{2}\|\mathbf{C}\|_F^2$ and $\ell(\cdot, \cdot)$ being the squared loss. The training procedure, as described in Ji et al. (2017), involves pre-training an antoencoder network without the $F(\mathbf{Z}, \mathbf{C})$ term. Such pretrained models for each of the three datasets are also provided alongside with their code. Then, the encoder and decoder networks of SEDSC are initialized by the pre-trained networks and all model parameters are trained via the Adam optimizer.

The implementation details of the three methods reported in Figure 2 and Table 1, namely Raw Data, Autoenc only and Full SEDSC, are as follows. For Raw Data, we solve the model in (1) with $\theta(\mathbf{C}) = \frac{1}{2}\|\mathbf{C}\|_F^2$ and $\lambda$ chosen in the set $\{0.5, 1, 2, 5, 10, 20, 50, 100, 200, 500\}$ that gives the highest averaged clustering accuracy over 10 independent trials. For Autoenc only, we use the pretrained encoder and decoder networks to initialize SEDSC, then freeze the encoder and decoder networks and use Adam to optimize the variable $\mathbf{C}$ only. The results for Full SEDSC are generated by running the code as it is. Finally, the same post-processing step is adopted for all three methods (i.e., we do not fine-tune it for Raw Data and Autoenc only).

## 6.2 EXPERIMENTS WITH SYNTHETIC DATA

In addition to the results shown in the main paper we additionally conduct similar experiments with synthetic data for the Instance Normalization and the Batch/Channel Normalization scheme.

**Details of experiments with real data.** For the Dataset and Batch/Channel normalization experiments we directly add a normalization operator to the encoder network which normalizes the output of the encoder such that the entire dataset has unit Frobenius norm (Dataset Normalization) or each row of the embedded dataset has unit norm (Batch/Channel Normalization) before passing to the self-expressive layer. For the Instance Normalization setting we add the regularization term proposed in Peng et al. (2017) with the form $\gamma_2 \sum_{i=1}^{N} (\mathbf{Z}_i^\top \mathbf{Z}_i - 1)^2$ to the objective in (3). We use regularization hyper-parameters $(\lambda, \gamma) = (10^{-4}, 2)$ for all cases and $\gamma_2 = 10^{-4}$ for the Instance Normalization case.

We first pretrain the autoencoder without the $F(\mathbf{Z}, \mathbf{C})$ term (i.e., $\gamma = 0$ and $\mathbf{C}$ fixed at $\mathbf{I}$), and we initialize the $\mathbf{C}$ matrix to be the solution to (1) using the $\mathbf{Z}$ embedding from the pretrained autoencoder. Following this we perform standard proximal gradient descent (Parikh & Boyd, 2013) on the full dataset, where we take a gradient descent step on all of the model parameters for the full objective excluding the $\theta(\mathbf{C})$ term, then we solve the proximal operator for $\theta(\mathbf{C})$. Figure 5 shows the results of this experiment, where we plot the original dataset along with the reconstructed output of

---

[3] https://github.com/panji1990/Deep-subspace-clustering-networks

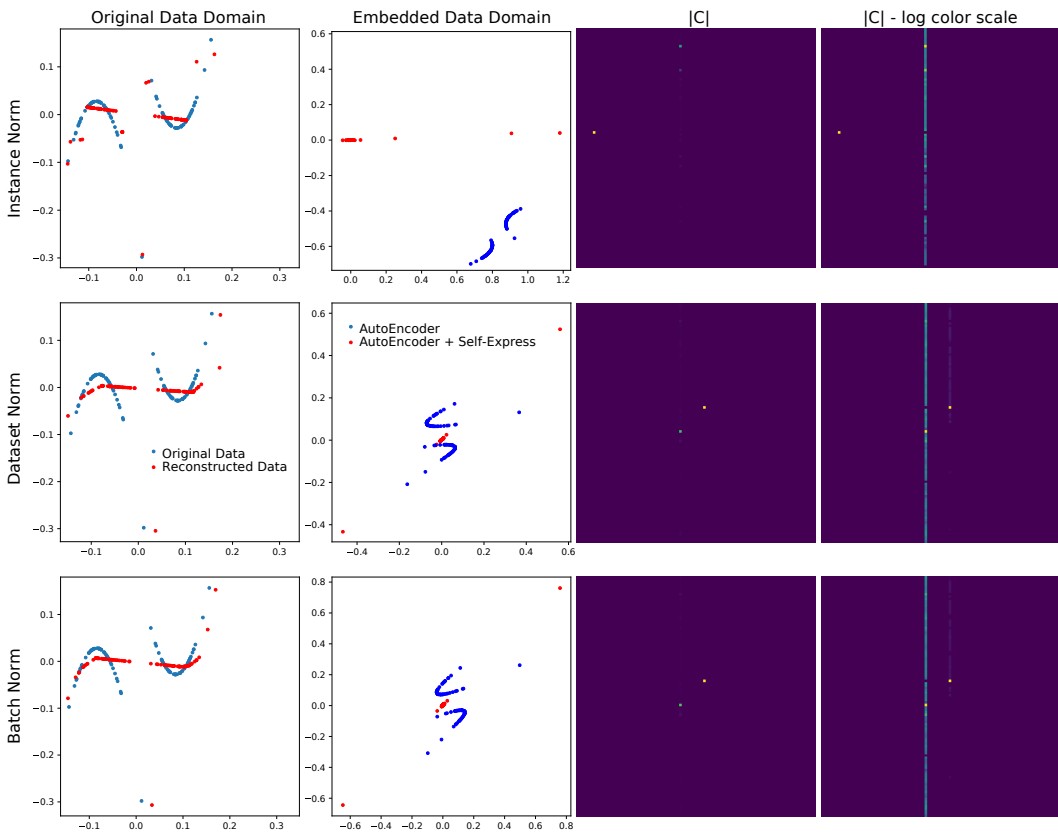

Figure 5: Showing results for the synthetic dataset for three normalization schemes (along the rows). Instance Normalization (top); Dataset Normalization (center); Batch/Channel Normalization (bottom). The columns are the same as described in the main paper. **(Left)** Original data points (Blue) and the data points at the output of the autoencoder when the full model (3) is used (Red). **(Center Left)** Data representation in the embedded domain when just the autoencoder is trained without the $F(\mathbf{Z}, \mathbf{C})$ term (Blue) and the full model is used (Red). **(Center Right)** The absolute value of the recovered $\mathbf{C}$ encoding matrix when trained with the full model. **(Right)** Same plot as the previous column but with a logarithmic color scale to visualize small entries.

the autoencoder (Left), the embedded representation after pretraining the autoencoder (Left Center-Blue) and after fully training the model (Left Center-Red), the absolute value of the final $\mathbf{C}$ matrix (Right Center), and the same plot with a logarithmic color scale to better visualize small entries (Right).

From Figure 5 one can see that our theoretical predictions are largely confirmed experimentally. Namely, first examining the Batch and Dataset normalization experiments one sees that when the full SEDSC model is trained the embedded representation is very close to as predicted by Theorem 2, with almost all of the embedded points (Left Center - Red points) being near the origin with the exception of two points, which are co-linear with each other. Likewise, the $\mathbf{C}$ matrix is dominated by two non-zero entries with the remaining non-zero entries only appearing on the log-scale color scale. Further, the Instance normalization experiment also largely confirms our theoretical predictions, where all the points are co-linear and largely identical copies of a single point.

