# OpenReview forum: "A Critique of Self-Expressive Deep Subspace Clustering"
_ICLR.cc/2021/Conference — ICLR 2021 Poster_

### Official Review · AnonReviewer1 · 2020-10-21
**Authors theoretically studied a class of self-expressive deep subspace clustering (SEDSC) methods, and found that autoencoder regularization formulation is typically ill-posed and the optimal embedding is still trivial after normalization is applied.**

**Rating:** 7
**Confidence:** 3

**Review:**

Pros:
1)	Authors theoretically studied a class of self-expression deep subspace clustering methods and found that the optimization problem is typically ill-posed.
2)	Various normalization approaches are studied, including dataset and batch/channel normalization, and instance normalization. However, even with these normalizations, the optimal embedded data geometry is still trivial in various ways.
3)	Authors conducted experiments on real and synthetic data to further verify the theoretical conclusions.

Cons:
1)	Although authors uncovered the ill-posed issue in existing SEDSC methods, it is more interesting to see how the ill-posed issue can be resolved or alleviated. A proper working solution can further improve the quality of this paper.
2)	The theoretical results seem mainly focusing on (3) with autoencoder regularization. Authors might want to specify the extension of the results to (2) in a more general form. Similarly, the discussion of various SEDSC methods cited in Section 1.1 in terms of the discovered results is important.
3)	Authors mentioned the solutions are optimal in many places of this paper. From the perspective of the nonlinear optimization problems, it is not proper to say “optimal solutions”.

I change my rating after looking at authors response.

---

> ### Author Response · Authors · 2020-11-18
> **Response to AnonReviewer1**
>
> Thank you for the feedback and noting the strengths of our work.  Here we reply to your specific comments.
>
> 1) It is more interesting to see how the issue can be resolved.
>
> Please see our reply to AnonReviewer2.
>
> 2) The results seem to focus on (3).  What about the more general form in (2)?
>
> First, we note that Prop 1 and 2 are the only results the specifically apply to (3), and even then there are still pieces of Prop 1 and 2 that apply to (2).  All of the other results in the paper apply to (2) as well.
>
> Further, we have focused on (3) because, to the best of our knowledge, a vast majority of the prior work is based on this model.  In particular, of the SEDSC works that we cite, Peng et al (2017) is the only work which is not based on (3), and that work is directly captured by our analysis of instance normalization in section 2.3.
>
> 3) It is not proper to say “optimal solutions”.
>
> We are not sure what the reviewer is referring to here.  In the problems we analyze there are infinitely many (non-unique) globally optimal solutions, and our theorems characterize the set of global minimizers.  If the reviewer could clarify what they are referring to we could attempt to address the point.

---

### Official Review · AnonReviewer4 · 2020-10-28

**Rating:** 7
**Confidence:** 4

**Review:**

This paper studies the flaws associated with extending subspace clustering methods to the nonlinear manifolds scenario. In particular, the authors demonstrate that the optimization problem solved due to the extension can be ill-posed and thus lead to solutions which are degenerate/trivial in nature. The paper also showed that the performance benefits often associated with the Self-Expressive Deep Subspace Clustering techniques are potentially due to post-processing steps and other factors rather than due to the efficacy of these methods themselves.

Overall the outline of the paper is good and I found the discussed problem informative. Few aspects were unclear to me (refer below).

1) In section 2, the authors discuss positively-homogeneous functions i.e. (leaky) Rectifier Linear Units and how it effects the statement in Proposition 1. I would like to understand how Proposition 1 relates to other activation functions i.e. the Hyperbolic Tangent (tanh) and Sigmoid functions for example. In general, given the latter activation functions are not positively-homogeneous and have disparate saturation/non-saturation regions, can we extend the analysis to these activation functions and make the theoretical parts of the paper more generic ?

2) In section 2.1, the authors talk about how Auto-encoders do not necessarily impose significant constraints on the geometric alignment of points. Are the authors aware of techniques like TopoAE and other state-of-the-art VAE/GAN and generative model variants which use some form of regularization to allow for topology modeling etc. and/or can produce results which achieve the above ? Did the authors consider this ?

3) How do the authors quantify which encoder-decoder architectures are "reasonably expressive" ? Does any constraint as part of the objective function hamper this or are they referring to more specific constraints ?

4) In sections 2.1 - 2.2, the authors mention that to remove trivial solutions the magnitude of the representations should be greater than some minimum value. What is this minimum value and how can we compute it efficiently ?

5) In section 2.2, the authors briefly talk about how optimal solutions to the optimization problem may exist which have zero mean. How often do the computed optimum values fall in this category or in general are degenerate or trivial ? My point is existence of degenerate solutions need not mean that our optimization process will actually end up with these degenerate/trivial solutions. We typically can add constraints (which act like a prior and factor in the objective) and thus push our final solution away from degenerate/trivial solutions.

I felt that the paper points out an important issue but if the authors could provide a general solution which helps ameliorate the issue (some general directions or even providing rudimentary results for one of the these directions) could have made the their contribution much more stronger. Overall I like the paper and the arguments made even though I have some inhibitions with regards to the scope of the contribution given the problem addressed is so specific.

---

> ### Author Response · Authors · 2020-11-18
> **Response to AnonReviewer4-Part1**
>
> We thank the reviewer for the constructive comments and for finding our discussion informative.  Here we address the specific questions that are raised.
>
> 1) How does Prop 1 relate to sigmoidal style non-linearities?
>
> Positive homogeneity is only a sufficient condition for the pathology described by Prop 1 to exist (and then only for the final claim of the statement) not a necessary condition.  We do not expect changing to a sigmoidal style non-linearity to change the behavior described by Prop 1.  Specifically, to reduce the overall objective it is beneficial to reduce the magnitude of the norm of the embedded representation.  As the entries of the embedded representation are reduced toward the origin, all of the sigmoidal-style non-linearities approach linear functions in a neighborhood around the origin, so we conjecture that the same pathology will result.
>
> Beyond this point, even if a sigmoidal style non-linearity did somehow address the pathology in Prop 1 (which we doubt), we do not consider this to be the key issue of the SEDSC model.  Indeed, as an example one can simply have the final non-linearity in the encoder be an explicit normalization operator to solve the Prop 1 issue, but as we show in the subsequent analysis this also typically results in trivial geometries in the embedded spaces.
>
> 2) Did the authors consider other auto-encoder approaches (e.g., TopoAE) to constraint the geometry of the embedded points?
>
> Indeed, there are a wide variety of approaches one could explore to attempt to correct the issues we describe and analyze regarding SEDSC, but this is not the point of our paper.  Rather, here we are pointing out through both theoretical analysis and experiments that there are significant flaws with the SEDSC model.  Further, to the best of our knowledge, none of the existing work on the SEDSC model has presented a solution which prevents trivial data geometries in the embedded space, and significantly more consideration needs to be given to the issues that we identify and analyze in this paper.
>
> 3) How is a “reasonably expressive” architecture defined?
>
> We do not make a formal definition of this or attempt to quantify the expressiveness of an architecture.  Rather, the point of the discussion in this section is to point out that if the encoder/decoder networks are infinitely expressive (i.e., can represent any possible function) then the optimal geometries in the embedded space will always approach the global optima described by the theorems in sections 2.2 and 2.3.  However, we also argue that one does not need an ‘infinitely expressive’ architecture to approach these optimal geometries in the embedded space.  As we formally show in Prop 2, simply having encoder/decoder networks with a single convolution channel each (which clearly cannot represent any possible function), for example, is sufficient for these trivial geometries to be the globally optimal solution.

---

> > ### Comment · AnonReviewer4 · 2020-11-18
> > **response**
> >
> > I would like to thank the authors for their response. I am currently going through the authors' response to my queries/concerns as well as those for the other reviewers and get back to them in case I have additional queries. I will update my score accordingly.

---

> ### Author Response · Authors · 2020-11-18
> **Response to AnonReviewer4-Part2**
>
> 4) What is the minimum value of the magnitude of embedded representation to avoid trivial solutions?
>
> To clarify, there is no minimum magnitude of the embedded representation which ensures non-trivial solutions.  The point we are making here is that if the first part of Prop 1 is satisfied then one can always produce an optimal solution (as least asymptotically) by simply doing the following:
>
> a) Train the autoencoder by itself, completely ignoring the F(Z,C) term (i.e., set \gamma = 0 and fix C as the identity matrix).
> b) Given an optimal solution for the autoencoder by itself, scale the weights of the encoder to drive the magnitude of the embedded representation towards 0 and scale the weights of the decoder to counteract the encoder scaling (i.e., scale the decoder weights to keep the final output of the autoencoder constant).
>
> Clearly, SEDSC provides no value for such solutions, since the F(Z,C) term becomes totally irrelevant, so one could simply train a generic autoencoder instead.  The comment about a minimum magnitude is simply to point out that if the magnitude of Z is somehow lower-bounded away from 0 then the above strategy cannot be used to produce a global minimum, but even in such cases trivial geometries result from the SEDSC model.  For example, the results of our Theorems 1-4 show that the global optima will have trivial geometries regardless of the value of $\tau$ in these Theorems.
>
> 5) How often do global minima have zero-mean?  How often are global minima trivial / how often does an optimization find a trivial solution?
>
> With regards to the zero-mean comment, this is simply to point out a connection with batch normalization.  This is referring to having a zero-mean for each row of Z, and from the result in Theorem 2, one can see that a zero-mean Z solution can always be constructed by having the two non-zero columns of Z have opposite sign.
>
> As for the question as to whether optimization will find a trivial solution, our theoretical analysis shows that globally optimal solutions to the SEDSC model will typically have trivial geometries.  It is potentially possible (although further analysis would be needed) that non-globally-optimal local minima might exist which have non-trivial geometries, but even if such spurious local minima did exist we consider a method/model which relies on converging to non-optimal solutions for good performance to be a fundamentally poor modeling philosophy.
>
> We also note that this is quite distinct from effects like implicit regularization in supervised learning.  In that regime there are infinitely many globally optimal solutions which can perfectly classify the training data, and the question becomes how a particular optimization method biases the model to choose a particular globally optimal solution which generalizes well to unseen data.  In the SEDSC setting we argue from our theoretical analysis that the only stationary points that will potentially have non-trivial geometries will be non-optimal by construction.  This would be somewhat akin to training a supervised network which can only generalize reasonably if you find local minima which correctly classify 75% of the training set, but if you find global minima which correctly classify 100% of the training data it cannot generalize.  One would likely not expect this to be a good strategy.
>
> Other Comments) How to fix these issues?  Somewhat limited scope of the paper.
>
> Please see our reply to AnonReviewer2.

---

### Official Review · AnonReviewer3 · 2020-10-28
**Valid points to consider when designing algorithms for deep self-expressive subspace clustering.**

**Rating:** 7
**Confidence:** 4

**Review:**

This paper critiques the commonly-used self-expressive cost function used to learn embeddings for deep subspace clustering. The authors point out that the empirical improvements obtained by deep self-expressive subspace clustering may be artifacts of post processing on the learned affinity matrix. They then theoretically characterize the optimal solutions to a variety of cost functions/normalization procedures used within the deep subspace clustering literature, showing that these encourage points to be mapped to a singleton set up to a sign change.

The theoretical contributions of this paper are significant, and the critique of deep self-expressive subspace clustering is timely and important. A survey of "shallow" subspace clustering methods shows that the alleged performance improvements obtained by deep subspace clustering typically amount to parameter tuning or post processing, as shallow methods often perform at least as well when properly tuned. The paper provides solid evidence that researchers should think more deeply when designing loss functions for unsupervised learning with neural networks. Empirical results are included that verify the theoretical contributions of this work.

---

> ### Author Response · Authors · 2020-11-18
> **Response to AnonReviewer3**
>
> We thank the reviewer for the positive comments about our work.

---

### Official Review · AnonReviewer2 · 2020-10-28
**A review for critique of SEDSC**

**Rating:** 7
**Confidence:** 3

**Review:**

Summary: The paper calls into question the significance of previous results on Self-Expressive Deep Subspace Clustering (SEDSC) models, which are touted as successful extensions of the linear subspace clustering (using the self-expressive property) to non-linear data structures. The authors present a set of theoretical results that indicate that the standard formulations of  SEDSC are generally ill-posed. Even with added regularizations, it is shown that such formulations could very well yield trivial geometries that are not conducive to successful subspace clustering.

Comments:
Although I have not fully checked the proof of the theoretical results, I believe this is a solid piece of work as it sheds light on shortcomings of SEDSC formulations using a rigorous theoretical approach. The authors verify their arguments using a good set of experiments. The findings also suggest that much of the claimed success of such models can in fact be attributed to post-processing of the encodings rather than the validity of the model.

- The paper has a limited scope as it raises concerns about an existing deep Subspace Clustering algorithm. I am not sure whether this algorithm is widely adopted and how significant it is -- the paper does also look at a class of similar formulations based on different regularizations. As such, I feel the paper addresses a somewhat limited audience and the impact of the work appears somewhat limited.

- Other than the limited scope, I do not see major weaknesses in this work, and I think the authors did a good job explaining the main ideas.

- The paper is hard to read for people who have not worked in closely related areas.

- The findings of this question beg the question -- which the paper does not attempt to answer -- as to whether there exist some other forms of regularizations for such models that would promote geometries of the embeddings that are conducive to successful clustering.

I believe this is a good paper worthy of being considered.

---

> ### Author Response · Authors · 2020-11-18
> **Response to AnonReviewer2**
>
> We thank the reviewer for the generally positive feedback and finding our results illuminating.  Here we comment on a few of the questions raised by the reviewer.
>
> 1) Paper is limited in scope/how wide-spread is the SEDSC approach?
>
> We cite ~20 papers in the manuscript which are based on SEDSC approaches, and we note that many groups appear to be continuing work based on SEDSC methods.  For example, even since our paper was initially submitted, additional papers based on SEDSC have appeared at NeurIPS (https://proceedings.neurips.cc/paper/2020/hash/753a043674f0193523abc1bbce678686-Abstract.html) and for review at ICLR (https://openreview.net/forum?id=WkKsWwxnAkt).  We believe it is essential to point out the potentially significant flaws with the SEDSC model that we discuss in this paper to ensure that ongoing and future work in this area takes these issues into consideration and avoids wasted effort on potentially flawed approaches.
>
> Further, while in this paper we focus on self-expressive deep subspace clustering (SEDSC) methods because it allows for a strong theoretical analysis, we conjecture that the issues we point out and analyze are not necessarily unique to SEDSC methods and could apply to other deep clustering approaches more generally given the potential for neural networks to have highly expressive mappings to the embedded space.
>
> 3) The paper is hard to read for those without a background in the area.
>
> We are sorry the reviewer had difficulty reading the manuscript, but we note that we included an extensive introduction section to present the development and background of prior work which informed the original SEDSC model for readers new to the area.  If the reviewer has specific suggestions for improving clarity we are happy to attempt to address them.
>
> 4) Is there some way to fix the SEDSC model to achieve successful clustering?
>
> While we are certainly interested in such questions, we note that the problem which SEDSC attempts to address is very challenging and one of the key problems in unsupervised learning (i.e., how to cluster data from complex manifold structures).  We consider a solution to this beyond the scope of the current paper.  However, that being said, it is easy to construct trivial special cases where the SEDSC model will at least not produce worse embedded data geometries than the original data geometry (as we argue it likely currently does).  For example, if the encoder/decoder are something like Resnet architectures trained with very high levels of weight decay regularization, then the learned parameters will be very small and the network functions will be dominated by the skip connections (i.e., the mappings will be close to the identity mapping).  This keeps the embedded geometry from being significantly worse than the original data geometry, but it also clearly does not likely provide much benefit over just clustering the original data either.

---

### Author Response · Authors · 2020-11-18
**Summary of changes in rebuttal submission.**

Here we have made a few minor changes from the original submission.

Specifically:

1) The initial proposition 1 had a minor error in the last claim of the statement, where a positive-homogeneity condition was required for both the encoding and decoding networks.  This has been corrected to only requiring a condition on the encoder, which is a weaker requirement than originally stated.

2) A minor change was made to proposition 2, where the second condition has been generalized to allow a norm of arbitrary magnitude $\tau > 0$ instead of 1.

3) Theorem 3 (Theorem 4 in the previous submission) has been stated in a more general way, as the result applies to any Schatten-p norm on C, not just the Frobenius and nuclear norms.  Likewise, the result has been moved from the supplement to the main paper.

---

### Decision · Program_Chairs · 2021-01-07
**Final Decision**

**Decision:**

Accept (Poster)

**Comment:**

The authors carefully study a class of unsupervised learning models called self-expressive deep subspace clustering (SEDSC) models,  which involve clustering data arising from mixtures of complex nonlinear manifolds. The main contribution is to show that the SEDSC formulation itself suffers from fundamental degeneracies, and that the experimental gains reported in the literature may be due to ad-hoc preprocessing.

The contributions are compelling, and all reviewers appreciated the paper. Despite the paper being of somewhat narrow focus, my belief is that negative results of this nature are useful and timely. I recommend an accept.